# RST1 and RIPR connect the cytosolic RNA exosome to the Ski complex in *Arabidopsis*

Heike Lange [1,5], Simon Y.A. Ndecky[1], Carlos Gomez-Diaz [1], David Pflieger[1], Nicolas Butel[2], Julie Zumsteg[1], Lauriane Kuhn [3], Christina Piermaria [3], Johana Chicher [3], Michael Christie[4], Ezgi S. Karaaslan [4], Patricia L.M. Lang [4], Detlef Weigel[4], Hervé Vaucheret[2], Philippe Hammann[3] & Dominique Gagliardi [1,5]

The RNA exosome is a key 3′—5′ exoribonuclease with an evolutionarily conserved structure and function. Its cytosolic functions require the co-factors SKI7 and the Ski complex. Here we demonstrate by co-purification experiments that the ARM-repeat protein RESURRECTION1 (RST1) and RST1 INTERACTING PROTEIN (RIPR) connect the cytosolic *Arabidopsis* RNA exosome to the Ski complex. *rst1* and *ripr* mutants accumulate RNA quality control siRNAs (rqc-siRNAs) produced by the post-transcriptional gene silencing (PTGS) machinery when mRNA degradation is compromised. The small RNA populations observed in *rst1* and *ripr* mutants are also detected in mutants lacking the RRP45B/CER7 core exosome subunit. Thus, molecular and genetic evidence supports a physical and functional link between RST1, RIPR and the RNA exosome. Our data reveal the existence of additional cytosolic exosome co-factors besides the known Ski subunits. RST1 is not restricted to plants, as homologues with a similar domain architecture but unknown function exist in animals, including humans.

[1] Institut de biologie moléculaire des plantes, CNRS, Université de Strasbourg, Strasbourg, France. [2] Institut Jean-Pierre Bourgin, INRA, AgroParisTech, CNRS, Université Paris-Saclay, Versailles, France. [3] Plateforme protéomique Strasbourg Esplanade FR1589 du CNRS, Université de Strasbourg, Strasbourg, France. [4] Max Planck Institute for Developmental Biology, Tübingen, Germany. [5] These authors jointly supervised this work: Heike Lange, Dominique Gagliardi. Correspondence and requests for materials should be addressed to H.L. (email: hlange@unistra.fr) or to D.G. (email: dominique.gagliardi@ibmp-cnrs.unistra.fr)

The RNA exosome provides all eukaryotic cells with a key 3′−5′ exoribonucleolytic activity that participates in the maturation of various non-coding RNAs and in the degradation of both non-coding and coding RNAs (reviewed in refs. [1–5]). The RNA exosome consists of a core complex composed of nine subunits (Exo9) to which the exoribonucleases RRP6 and DIS3/RRP44 differentially associate within the nucleolus, nucleoplasm or cytosol[6–8]. Whilst the overall structure and function of the RNA exosome is conserved, both the composition and enzymatic activities of exosome complexes vary among organisms. For example, most non-plant Exo9s including those in yeast and human are catalytically inactive[9], whereas plant Exo9s have retained a phosphorolytic activity originating from its prokaryotic ancestor[10]. This unique phosphorolytic activity of plant Exo9 acts in combination with the hydrolytic activities provided by RRP6 and DIS3[10]. Another exception among RNA exosomes is the association of human Exo9 with functionally distinct DIS3L and DIS3 proteins, only the latter of which is conserved in yeast and plants[6,11].

In all eukaryotes investigated, the catalytic activities of the RNA exosome are modulated by cofactors termed activator–adapter or exosome targeting complexes. These complexes aid in the recognition of specific types of RNA substrates and couple exosome-mediated degradation to cellular processes, such as ribosome biogenesis or mitosis[12–19]. All exosome targeting complexes that have been characterised to date contain an RNA helicase from the MTR4/SKI2 family as a central component. In addition, exosome targeting complexes typically comprise RNA-binding proteins, non-canonical poly(A) polymerases or factors mediating protein–protein interactions. Most exosome targeting complexes described to date are nuclear. They include the TRAMP (TRF4-AIR1-MTR4 polyadenylation) complexes[20–22] in both baker's yeast and humans, the human PAXT (polyA tail exosome targeting) complex[23], the NEXT (nuclear exosome targeting) complexes that differ slightly in humans and plants[24,25] and the MTREC (Mtr4-like 1 (Mtl1)-Red1-core) complex in fission yeast[26,27]. These MTR4 containing complexes assist the exosome in nuclear RNA surveillance by targeting various RNA substrates, including precursors of ribosomal and other non-coding RNAs, spurious transcripts generated by pervasive transcription and untimely, superfluous or misprocessed mRNAs[21,24,25,27–32]. In remarkable contrast to the diversity of nuclear exosome cofactors, a single conserved protein complex, the Superkiller (Ski) complex, is known to assist the exosome in the cytosol. The Ski complex consists of the MTR4-related RNA helicase SKI2, the tetratricopeptide-repeat protein SKI3 and two copies of the WD40-repeat protein SKI8[33–35]. Association of the Ski complex with the exosome core complex requires an additional protein, SKI7[36]. Recent data revealed the functional conservation of SKI7 across eukaryotes[37]. In mammals and plants, SKI7 is produced by alternative splicing from a single locus that encodes also the HBS1 protein[37–39]. HBS1 functions together with the G-protein Dom34/PELOTA in No-Stop decay by releasing ribosomes stalled on RNAs lacking a stop codon[40–43]. In the yeast *Saccharomyces cerevisiae*, Ski7 and Hbs1 are closely related paralogs. Therefore, it was for long inferred that yeast Ski7 mediates the association of the exosome with the ribosome[34,44]. Recent data now challenge this view by showing that the Ski2–Ski3–Ski8 complex can directly bind to ribosomes, while Ski7 is associated with Exo9[35,45].

The Ski complex is conserved in *Arabidopsis thaliana*[46], but its physical association with the exosome core has not been investigated yet. An initial experiment to affinity-capture factors associated with the *Arabidopsis* exosome identified the homologue of DIS3 and two nuclear RNA helicases, AtMTR4 and its closely related homologue HEN2[25]. In addition, *Arabidopsis*

Exo9 systematically co-purified with a 1840 amino acid ARM-repeat protein of unknown molecular function named RESUR-RECTION1 (RST1)[25]. RST1 was originally identified in a genetic screen for factors involved in the biosynthesis of epicuticular waxes[47]. Epicuticular waxes are a protective layer of aliphatic very long-chain (VLC) hydrocarbons that cover the outer surface of land plants[48]. *rst1* mutants have less wax on floral stems than wild-type plants, and ~70% of the seeds produced by *rst1* mutants are shrunken due to aborted embryogenesis[47]. The molecular function of RST1 remains unknown. Interestingly, one of the two RRP45 exosome core subunits encoded in the *Arabidopsis* genome, named RRP45B or CER7 (for ECERIFERUM 7) was also identified in a genetic screen aimed at identifying enzymes or regulators of wax biosynthesis[49]. The wax-deficient phenotype of *rrp45b/cer7* mutants (*cer7* from now on) is suppressed by mutations in genes encoding RNA silencing factors, such as RDR1, RDR6, AGO1, SGS3 and DCL4[50,51]. This and the identification of small RNAs accumulating in *cer7* mutants revealed that the wax deficiency observed in *cer7* plants is due to post-transcriptional silencing of *CER3* mRNAs[50,51], encoding a protein that together with the aldehyde decarbonylase CER1 catalyses the synthesis of VLC alkanes from VLC acyl-CoAs[52,53]. These results demonstrated that the RNA exosome contributes to the degradation of the *CER3* mRNA and that the wax-deficient phenotype of *cer7* mutants is a consequence of the established link between RNA degradation and silencing pathways[50,51,54]. Indeed, in plants, the elimination of degradation intermediates such as uncapped or RISC-cleaved mRNAs by 3′−5′ and 5′−3′ exoribonucleases prevents that they trigger post-transcriptional silencing (PTGS)[55–60], a mechanism required for the destruction of non-self RNAs originating from viruses or transgenes.

Here, we demonstrate by multiple reciprocal co-purification assays coupled to mass spectrometry analyses that the *Arabidopsis* exosome core complex Exo9 associates with the ARM-repeat protein RST1, SKI7, another protein that we named RIPR (for RST1 interacting protein) and the Ski complex. Our data show that RST1 and RIPR suppress the silencing of transgenes as well as the production of secondary siRNAs from endogenous exosome targets such as RISC-cleaved transcripts and certain endogenous mRNAs which are prone to PTGS. Those mRNAs include the *CER3* mRNAs explaining that the *rst1* and *ripr* mutants share the *cer7* wax-deficiency phenotype. Taken together, our biochemical and genetic data establish RST1 and RIPR as cofactors of the cytoplasmic exosome and the Ski complex in plants.

## Results

**Wax deficiency in *rst1* mutants is caused by *CER3* silencing**. To investigate whether the wax deficiency of *rst1* plants is linked to compromised degradation of the *CER3* mRNA as reported in *cer7* mutants[49], we compared the stems of wild-type and mutant plants grown under identical conditions. Due to the light reflecting properties of the wax crystals that cover the outer cuticle, stems of wild-type *Arabidopsis* plants appear whitish (or bluish in cold light, Fig. 1a). Consistent with previous reports, plants lacking the exosome core subunit RRP45A have whitish stems signifying intact wax biosynthesis[49]. By contrast, plants with T-DNAs inserted in the *RRP45B/CER7* (*AT3G60500*) and *RST1* (*AT3G27670*) loci have glossy green stems indicating wax deficiency[47,49] (Fig. 1a). Gas chromatography followed by mass spectrometry (GC-MS) analysis of extracts obtained from the stem surface confirmed that the amounts of the VLC derivatives nonacosane, 15-nonacosanone and 1-octacosanol, three major components of epicuticular stem wax in *Arabidopsis*, were similarly reduced in *cer7* and in *rst1* mutants (Fig. 1b). Ectopic

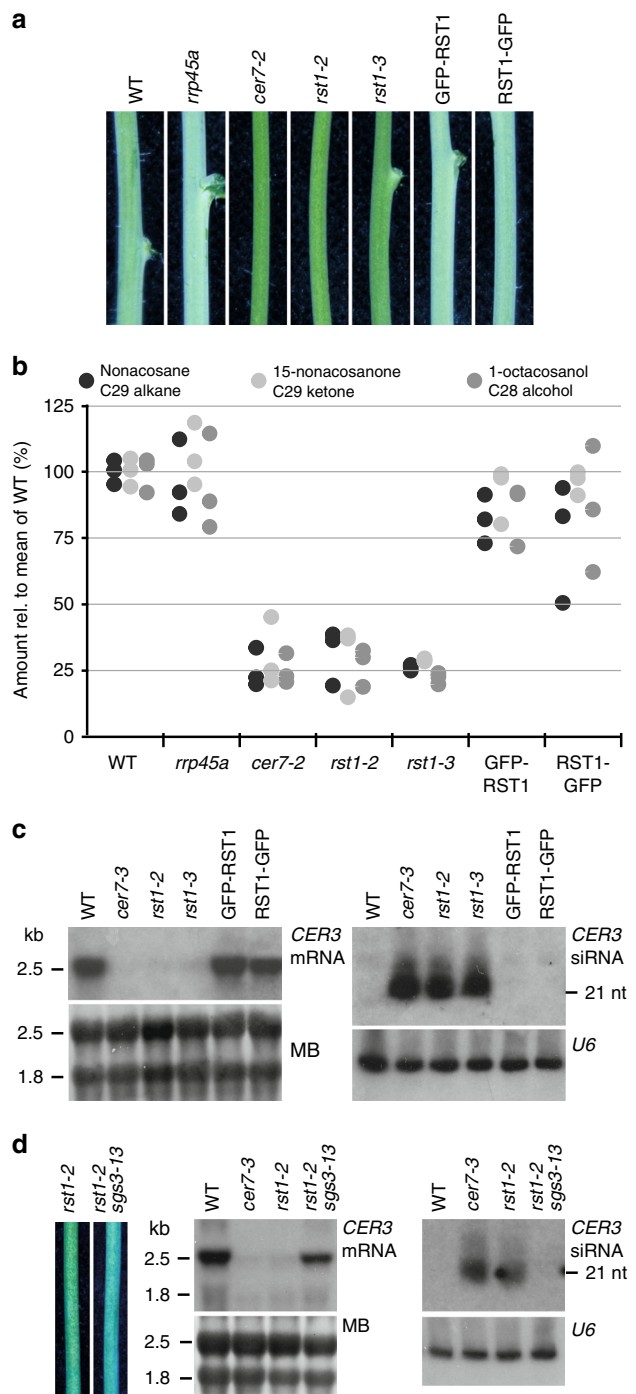

**Fig. 1** The wax-deficient phenotype of *rst1* mutants is caused by silencing of the *CER3* gene. **a** Inflorescence stems of *Arabidopsis* plants of the indicated genotypes. GFP-RST1 and RST1-GFP are *rst1-3* plants expressing RST1 fused to GFP. The whitish appearance of wild-type stems is due to a layer of the epicuticular wax deposited on the stem surface, while wax-deficient stems appear green and glossy. To better visualise the difference between normal and glossy stems, the white balance of the photograph was set to cold light (3800 K), which accounts for the bluish appearance of the picture. **b** Relative amounts of major stem wax compounds extracted from *Arabidopsis* stem sections. **c** Levels of *CER3* mRNA and *CER3*-derived siRNAs in WT, *cer7* and *rst1* mutants. The total RNA extracted from stem samples of the indicated genotypes was separated by denaturing agarose (left) or polyacrylamide (right) electrophoresis, transferred to membranes and hybridised with a probe specific to *CER3*. The methylene blue (MB) stained membrane and hybridisation with a probe specific to U6 snRNA are shown as loading controls, respectively. **d** Mutating *SGS3* restores the wax phenotype of *rst1* mutants. Stem sections from *rst1-2* and *rst1-2 sgs3-13* plants are shown on the left. RNA blots show full-length *CER3* mRNA (mid), and *CER3*-derived siRNAs (right) in RNA samples extracted from stems of the indicated genotypes. The source data are available at [https://doi.org/10.6084/m9.figshare.c.4483406].

phenotype of *rst1* mutants is caused by silencing of the *CER3* mRNA, as reported for *cer7* mutants.

**RST1 is a suppressor of transgene silencing**. Two independent genetic screens identified *rst1* point mutations as suppressors of silencing. The first screen aimed to identify mutations suppressing the phenotype of *MIM156* plants. *MIM156* plants express an artificial non-coding RNA with an uncleavable miRNA 156 recognition site[61]. Ectopic expression of this miRNA target mimicry (MIM) construct reduces both levels and activity of endogenous miR156 and leads to a characteristic phenotype with spoon-shaped cotyledons, prematurely serrated rosette leaves and a reduced leaf initiation rate during vegetative growth (Fig. 2a). *MIM156 rst1-4* plants were recovered from an EMS-treated population of *MIM156* plants that had been visually screened for restoration of normal growth and development. *MIM156 rst1-4* plants display the spoon-shaped cotyledons of the parental line, but wild-type-like leaf initiation rates and rosette leaf serration. We mapped the suppressor mutation by whole-genome sequencing to the *RST1* gene and specifically the G3118A mutation causing a G706D amino acid change (Fig. 2b). Expressing a genomic *RST1* construct in *MIM156 rst1-4* plants restored the *MIM156* phenotype confirming that the *rst1-4* mutation was responsible for the suppressor effect (Fig. 2a). We then tested the accumulation of the *CER3* mRNA and *CER3*-derived siRNAs in this novel *rst1* allele. As compared with the T-DNA insertion alleles *rst1-2* and *rst1-3*, *rst1-4* mutants had residual levels of the full-length *CER3* mRNA and lower levels of *CER3*-derived siRNAs, indicating that *rst1-4* is a weak allele (Fig. 2c). Next, we analysed the accumulation of both the full-length *MIM156* non-coding RNA and *MIM156*-derived siRNAs by RNA blots. This experiment revealed low levels of *MIM156*-derived siRNAs in the parental *MIM156* line indicating that the *MIM156* transcript is spontaneously targeted by PTGS, as often observed with highly expressed transgenes. Compared with the parental line, *MIM156 rst1-4* plants had reduced levels of the full-length *MIM156* transcript, but accumulated increased levels of *MIM156*-derived siRNAs (Fig. 2d). The increased accumulation of these siRNAs in the *MIM156 rst1-4* suggests that RST1 restricts the production of *MIM156*-derived siRNAs, which prevents complete destruction of the full-length transcript by PTGS.

expression of RST1 fused to GFP in *rst1-3* plants restored the biosynthesis of nonacosane, 15-nonacosanone and 1-octacosanol and resulted in wild-type-like whitish stems (Fig. 1a, b). Previous studies established that the wax deficiency of *cer7* mutants is due to post-transcriptional silencing of the mRNA encoding *CER3*[49–51,54], a subunit of a VLC alkane-forming complex[52,53]. Indeed, RNA blots revealed a severe reduction of the *CER3* mRNA and an accumulation of *CER3*-derived small RNAs in both *cer7* and *rst1* mutants (Fig. 1c). Mutating the PTGS factor SUPPRESSOR OF GENE SILENCING 3 (SGS3) in *rst1-2* plants abolished the production of *CER3*-derived small RNAs, restored wild-type levels of the *CER3* mRNA and allowed the production of epicuticular wax as demonstrated in the *rst1 sgs3* double mutant (Fig. 1d). These results show that the wax-deficient

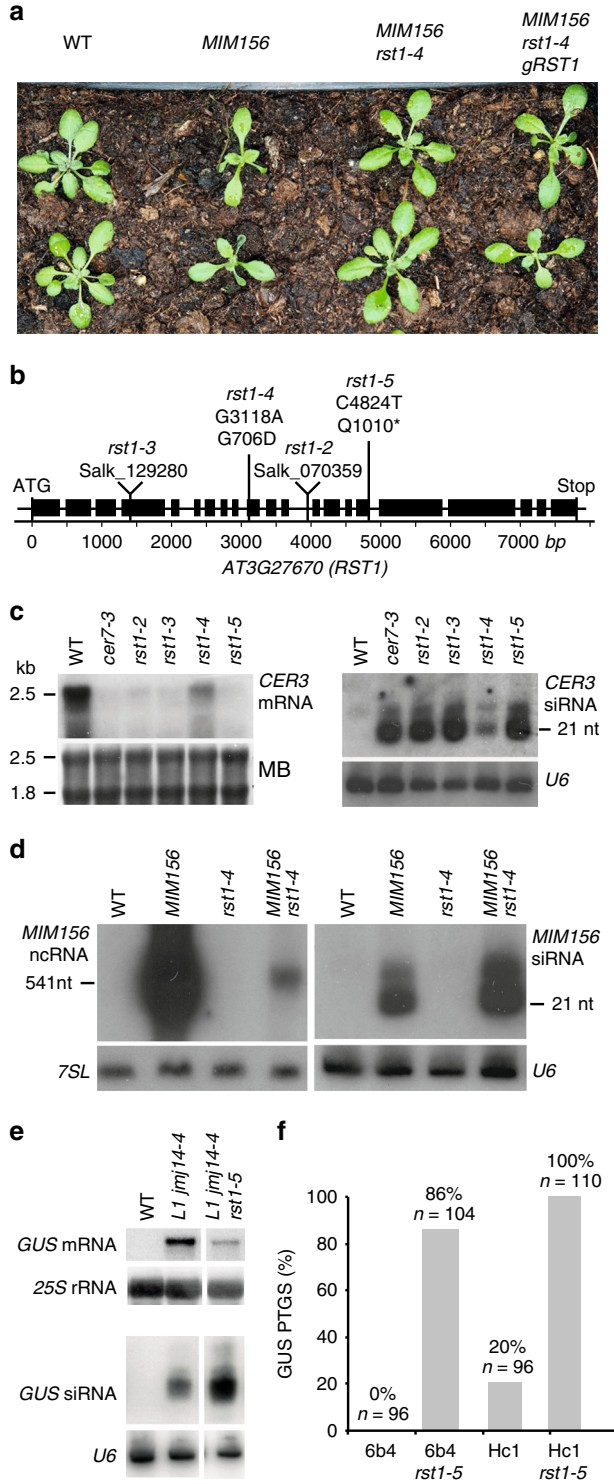

**Fig. 2** RST1 suppresses silencing of transgenes. **a** The *rst1-4* mutation suppresses the developmental phenotype induced by a *MIM156* transgene. **b** Diagram of the *AT3G27670* gene encoding the RST1 protein. Boxes represent exons, lines represent introns. Triangles indicate the position of the T-DNA insertions in *rst1-2* and *rst1-3* lines. Vertical lines indicate the point mutations in *rst1-4* and *rst1-5*. **c** *rst1-4* is a weak allele. Accumulation of the *CER3* mRNA (left) and *CER3*-derived siRNAs (right) in wild-type (WT), *cer7-3* and the four *rst1* alleles used in this study shown by RNA blots hybridised with a probe specific to CER3. The methylene blue stain of the membrane (MB) and hybridisation to U6 snRNA are shown as loading controls. **d** RNA blots showing the accumulation of the full-length MIM156 ncRNA and MIM156-derived siRNAs visualised by hybridisation with a probe specific to the IPS1 backbone of the MIM156 transgene. 7SL RNA and U6 snRNA are shown as loading controls. **e** RNA blots showing that the *rst1-5* mutation results in reduced levels of the *GUS* mRNA and increased levels of *GUS*-derived siRNAs in the L1 *jmj14-4* background. 25S rRNA and U6 snRNA are shown as loading controls. **f** The *rst1-5* mutation increases S-PTGS frequency in both 6b4 and Hc1 reporter lines. The barplot shows the proportion of plants with silenced *GUS* expression in the indicated genotypes. The source data are available at [https://doi.org/10.6084/m9.figshare.c.4483406]

siRNA similar to *rst1-2* and *rst1-3* (Fig. 2c). To further demonstrate that *rst1-5* enhances PTGS, we introduced the *rst1-5* mutation into the well-established reporter lines *6b4* and *Hc1*[57] (Fig. 2f). These lines harbour the same *35S:GUS* transgene as the *L1 jmj14-4* line, but inserted at different locations in the *Arabidopsis* genome. In a wild-type background, line *6b4* does not trigger sense transgene PTGS (S-PTGS), while line *Hc1* triggers S-PTGS in 20% of the population (Fig. 2f). In genetic backgrounds having impaired RNA degradation, both Hc1 and 6b4 lines trigger S-PTGS at increased frequencies, which provides a quantitative readout[25,55–57,59,63]. 86% of the 6b4 *rst1-5* plants and 100% of the Hc1 *rst1-5* plants triggered silencing of the *35S:GUS* reporter (Fig. 2f). This result confirmed that RST1 functions as a suppressor of S-PTGS comparable with other proteins involved in RNA degradation[25,55–57,59,63]. Of note, the role of RST1 as S-PTGS suppressor is also supported by an independent study published during the reviewing process of our paper[64].

**RST1 co-purifies with the exosome, SKI7 and RIPR**. To examine the intracellular distribution of RST1, we used a *rst1-3* mutant line expressing RST1 proteins fused to GFP at its N- or C-terminus. Both fusion proteins were functional as they rescued the wax deficiency of *rst1-3* (Fig. 1), and showed a diffuse cytoplasmic distribution in root cells of stable *Arabidopsis* transformants similar to the cytoplasmic marker protein PAB2 (Fig. 3). Therefore, we conclude that RST1 is a cytoplasmic protein. The diffuse cytoplasmic localisation of RTS1 is in agreement with a recent independent study[65]. In our previous experiments, RST1 co-purified with Exo9 using the core exosome subunit RRP41 as bait[25]. To verify the association of RST1 with Exo9, we used GFP-RST1 or RST1-GFP as baits for immunoprecipitation (IP) experiments followed by LC-MS/MS analyses (15 IPs). Indeed, amongst the proteins that were enriched in RST1 IPs were the nine canonical subunits of Exo9: CSL4, MTR3, RRP4, RRP40A, RRP41, RRP42, RRP43, RRP45B/CER7 and RRP46 (Fig. 4a; Supplementary Data 1). By contrast, we did not detect HEN2 or MTR4, the two main cofactors of nucleoplasmic or nucleolar exosomes, respectively. Noteworthy, RST1 co-purified with the cytoplasmic protein encoded by *AT5G10630*, the mRNA of which is alternatively spliced to produce either HBS1 or SKI7 proteins. As compared with the *HBS1* mRNA, the *SKI7* mRNA contains an additional exon encoding the putative exosome interaction

The second screen directly aimed at identifying factors affecting the post-transcriptional silencing of the *35Sprom:GUS* transgene in the reporter line *L1 jmj14-4*[62]. This screen identified *rst1-5*, a C4824T mutation in *RST1* resulting in a truncation of the RST1 protein (Q1010*) (Fig. 2b). Compared with the *L1 jmj14-4* parental line, *L1 jmj14-4 rst1-5* plants had decreased levels of *GUS* mRNA and increased levels of *GUS*-derived siRNA (Fig. 2e). This result resembled the effects of *rst1-4* on the accumulation of *MIM156* transcript and *MIM156*-derived siRNAs (Fig. 2d). Backcrossing *L1 jmj14-4 rst1-5* to wild-type yielded *rst1-5* plants, which showed a pronounced accumulation of *CER3*-derived

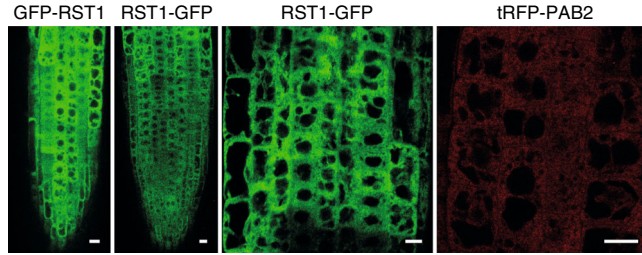

**Fig. 3** RST1 is a cytosolic protein. Confocal microscopy of root tips from *Arabidopsis rst1-3* plants expressing GFP-RST1 and RST1-GFP fusion proteins under the control of the constitutive *UBIQUITIN 10* promoter. Poly(A)-binding protein 2 at its N-terminus fused to tRFP (tRFP-PAB2) and expressed under the control of its own promoter is shown as cytosolic marker. Scale bars are 10 µm. The source data are available at [https://doi.org/10.6084/m9.figshare.c.4483406]

domain of the *Arabidopsis* SKI7 protein[37,39]. Inspection of the peptides detected in the RST1 IP revealed the presence of peptides specific to the SKI7 splice isoform (Supplementary Fig. 1). This and the fact that SKI7 proteins are bound to yeast and human exosome complexes indicate that the *AT5G10630* gene product which co-purified with RST1 is indeed SKI7 rather than HBS1. The three core proteins of the cytoplasmic Ski complex SKI2, SKI3 and SKI8 were not significantly enriched (Fig. 4a). Together, these IP results support the exclusively cytoplasmic localisation of RST1 and confirm its interaction with the Exo9 core complex. Furthermore, a protein of unknown function encoded by *AT5G44150* and that we termed RIPR for RST1 INTERACTING PROTEIN was the most enriched protein in all RST1 IPs (Fig. 4a).

Because our previous purifications of *Arabidopsis* exosome complexes with tagged RRP41 as bait were analysed using an older and less sensitive mass spectrometer[25], we repeated the experiment using the same experimental settings as we used for the RST1 IPs (6 IPs, Fig. 4b; Supplementary Data 2). This new experiment confirmed the previously reported co-purification of the conserved exoribonuclease RRP44 and the two nuclear RNA helicases MTR4 and HEN2 with Exo9 and reproduced the co-purification of RST1. In addition, the new experiment revealed a significant enrichment of both RIPR and SKI7 in the RRP41 IPs.

Peptides specific to the alternative subunit RRP45A were present in the RRP41 IPs, but absent when RST1 was used as bait, suggesting that RST1 may preferentially interact with CER7 (aka RRP45B)-containing exosome complexes. To test this hypothesis, we stably expressed GFP-tagged RRP45A and RRP45B/CER7 in *Arabidopsis*. Indeed, both RST1 and SKI7 were significantly enriched with CER7 as bait (Fig. 5; Supplementary Data 3). RRP45A co-purified with the nucleoplasmic RNA helicase HEN2, which confirmed the association of HEN2 with Exo9-RRP45A previously observed in the reciprocal IP[27]. The high number of experiments (15 IPs) was also sufficient to detect the exoribonuclease RRP44, which is consistently poorly enriched in IPs of the *Arabidopsis* Exo9[10,25] (also seen in Fig. 4). We cannot formally rule out that the failure to detect RST1 and SKI7 with RRP45A-GFP is due to technical reasons. However, our data suggest that RST1 and SKI7 are principally associated with RRP45B/CER7. The difference in the interactome of CER7 and RRP45A cannot be explained by different intracellular localisation of the baits, because both CER7-GFP and RRP45A-GFP were present in the nuclear and cytoplasmic compartments (Supplementary Fig. 2).

In order to confirm the physical association of RIPR with RST1, we used RIPR with GFP-tags at either the N-terminal or the C-terminal ends as bait in co-purification experiments (4 IPs, Fig. 6; Supplementary Data 4). RST1 was the most enriched protein in RIPR IPs. The nine subunits of the exosome were also

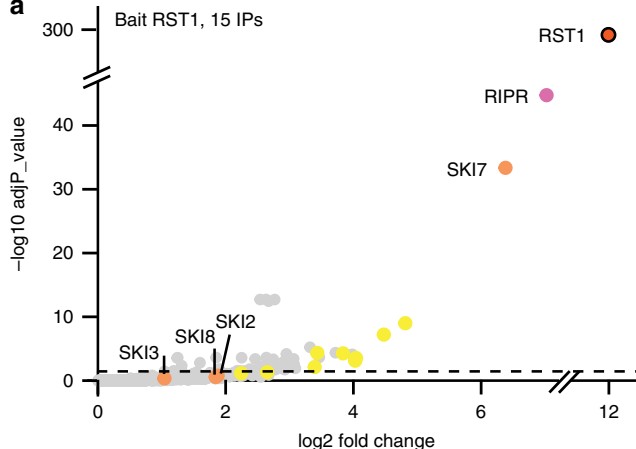

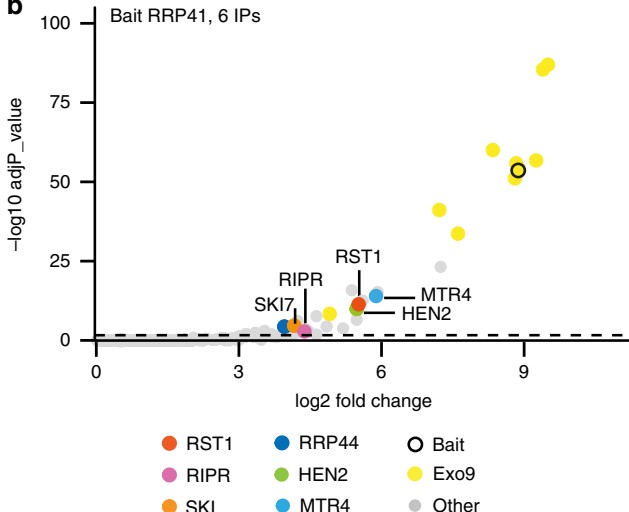

**Fig. 4** RST1 co-purifies with the exosome, SKI7 and RIPR. Volcano plots show the enrichment of proteins co-purified with GFP-tagged RST1 (**a**) or RRP41 (**b**) as compared with control IPs. Y- and X-axis display adjusted *p*-values and fold changes, respectively. The dashed line indicates the threshold above which proteins are significantly enriched (adjP < 0.05). The source data are available in Supplementary Data 1 and 2

detected, but were less enriched than in the IPs with RST1 as bait (compare Figs. 4, 6). By contrast, SKI7 as well as the three components of the Ski complex SKI2, SKI3 and SKI8 were amongst the most significantly enriched proteins co-purifying with RIPR (Fig. 6).

Altogether, the multiple reciprocal IPs confirm the interaction of RST1 with CER7-containing exosome core complexes and identify SKI7 and RIPR as additional binding partners of both RST1 and Exo9. Furthermore, our data indicate that RST1 binds Exo9 and SKI7, while RIPR binds to RST1-SKI7 and the Ski complex.

**Loss of RIPR function phenocopies *rst1* mutants.** RIPR is conserved amongst flowering plants but has no clear sequence homologues in mosses, green algae or outside of the green lineage. RIPR is a 356 amino acid protein that lacks obvious functional domains and motifs or sequence homologies to known proteins. Confocal microscopy of *Arabidopsis* roots stably expressing RIPR-GFP fusion proteins revealed a diffuse cytoplasmic distribution (Fig. 7a) similar to the intracellular distribution of RST1

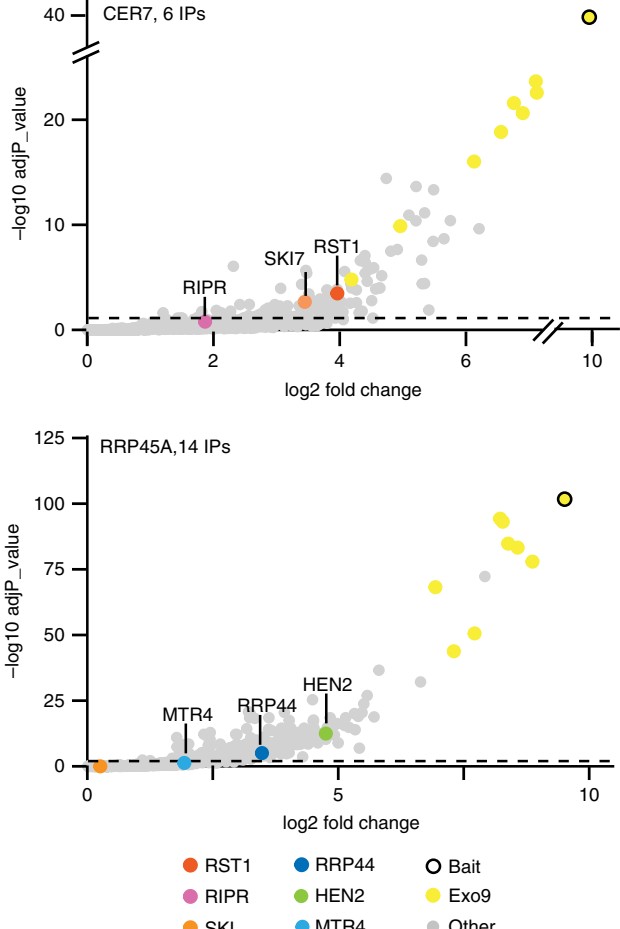

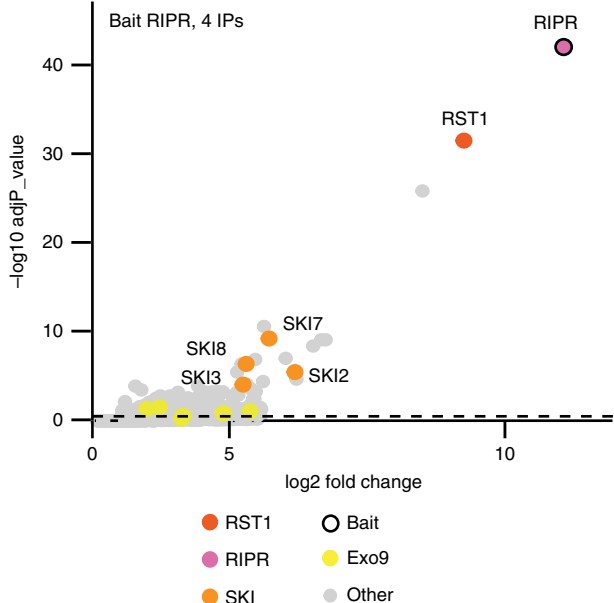

**Fig. 6** RIPR co-immunoprecipitates RST1, SKI7 and the Ski complex. The Volcano plot shows the enrichment of proteins co-purified with GFP-tagged RIPR as compared with control IPs. Y- and X-axis display adjusted p-value and fold change, respectively. The dashed line indicates the threshold above which proteins are significantly enriched. The source data are available in Supplementary Data 4

**Fig. 5** RST1 and RIPR are bound to CER7-containing exosomes. Volcano plots show the enrichment of proteins co-purified with GFP-tagged RRP45B/CER7 (**a**) or RRP45A (**b**) as compared with control IPs. Y- and X-axis display adjusted p-value and fold change, respectively. The dashed line indicates the threshold above which proteins are significantly enriched (adjP< 0.05). The source data are available in Supplementary Data 3

(Fig. 3). Because T-DNA insertion mutants in the *AT5G44150* locus were not available in *Arabidopsis* stock centres, we obtained two independent mutants using a CRISPR-Cas9 strategy. The mutants, named *ripr(insT)* and *ripr(insC)*, had single T or C nucleotides inserted at position 179, creating premature stop codons 60 and 65 amino acids after the start codon, respectively (Fig. 7b, c; Supplementary Fig. 3). Interestingly, *ripr(insT)* and *ripr(insC)* mutant plants have glossy green stems resembling the stems of *rst1* and *cer7* plants (Fig. 7d). Moreover, about 70% of the seeds produced by *ripr(insT)* and *ripr(insC)* plants were shrunken, similar to the proportion of unviable seeds produced by *rst1-2*, *rst1-3* or *cer7-3* plants (Fig. 7e; Supplementary Fig. 4)[47]. RNA blots confirmed that the wax-deficient phenotype of *ripr (insT)* and *ripr(insC)* is due to the accumulation of *CER3*-derived small RNAs and silencing of the *CER3* mRNA (Fig. 7f). Those results demonstrate that RIPR can, like RST1, suppress the production of small RNAs from the *CER3* locus. Finally, 56/56 Hc1 *ripr(insT)* plants triggered silencing of the GUS PTGS reporter (Fig. 7g), demonstrating that RIPR also supresses transgene silencing. Taken together, loss of RIPR induced very similar physiological and molecular phenotypes as loss of RST1 or the exosome subunit CER7.

**mRNA-derived small RNAs accumulate in *cer7*, *ripr* and *rst1*.** The association of RST1 and RIPR with the exosome complex and the fact that loss of RST1 or RIPR phenocopies the *cer7* mutation suggested that both RST1 and RIPR are involved in the exosome-mediated degradation of the *CER3* mRNA before it can become a template for the production of *CER3*-derived small RNAs. To identify other common targets of RST1, RIPR and the exosome complex, we analysed small RNA libraries prepared from wild-type plants and from *cer7*, *rst1* and *ripr* mutants (Fig. 8a). This analysis identified more than 300 mRNAs that gave rise to small RNAs in *cer7* (Supplementary Data 5), including five of the six mRNAs that were previously shown to undergo silencing in absence of the RRP45B/CER7 exosome subunit[51]. Many of the loci that generate small RNAs in *cer7* mutants have previously been shown to produce siRNAs in *ski2* single mutants, *ski2 xrn4* double mutants or in the decapping mutants *dcp2* and *vcs*[57,58,60], and/or are known or predicted targets of miRNAs (Supplementary Data 5). Because these siRNAs are produced from protein-coding genes in RNA degradation mutants, they have been termed ct-siRNAs (coding-transcript siRNAs) or rqc-siRNAs (RNA quality control siRNAs)[57,58,60]. About one-third of the loci that generated rqc-siRNA in *cer7* mutants produced significant amounts of rqc-siRNAs in *rst1* and *ripr* mutants as well, while only very few loci were specifically observed in only *ripr* or *rst1* (Fig. 8b). The observation of quasi identical populations of small RNAs in *rst1* and *ripr* and the fact that almost all loci affected by *rst1* or *ripr* are also affected in *cer7* strongly support the conclusion that RST1 and RIPR are required for the degradation of at least a subset of cytoplasmic exosome targets.

## Discussion

This study identifies RST1 and RIPR as two previously unknown cofactors which support the function of the cytoplasmic RNA exosome in *Arabidopsis*. Three lines of evidence back our conclusion. Firstly, RST1 and RIPR are physically associated with the

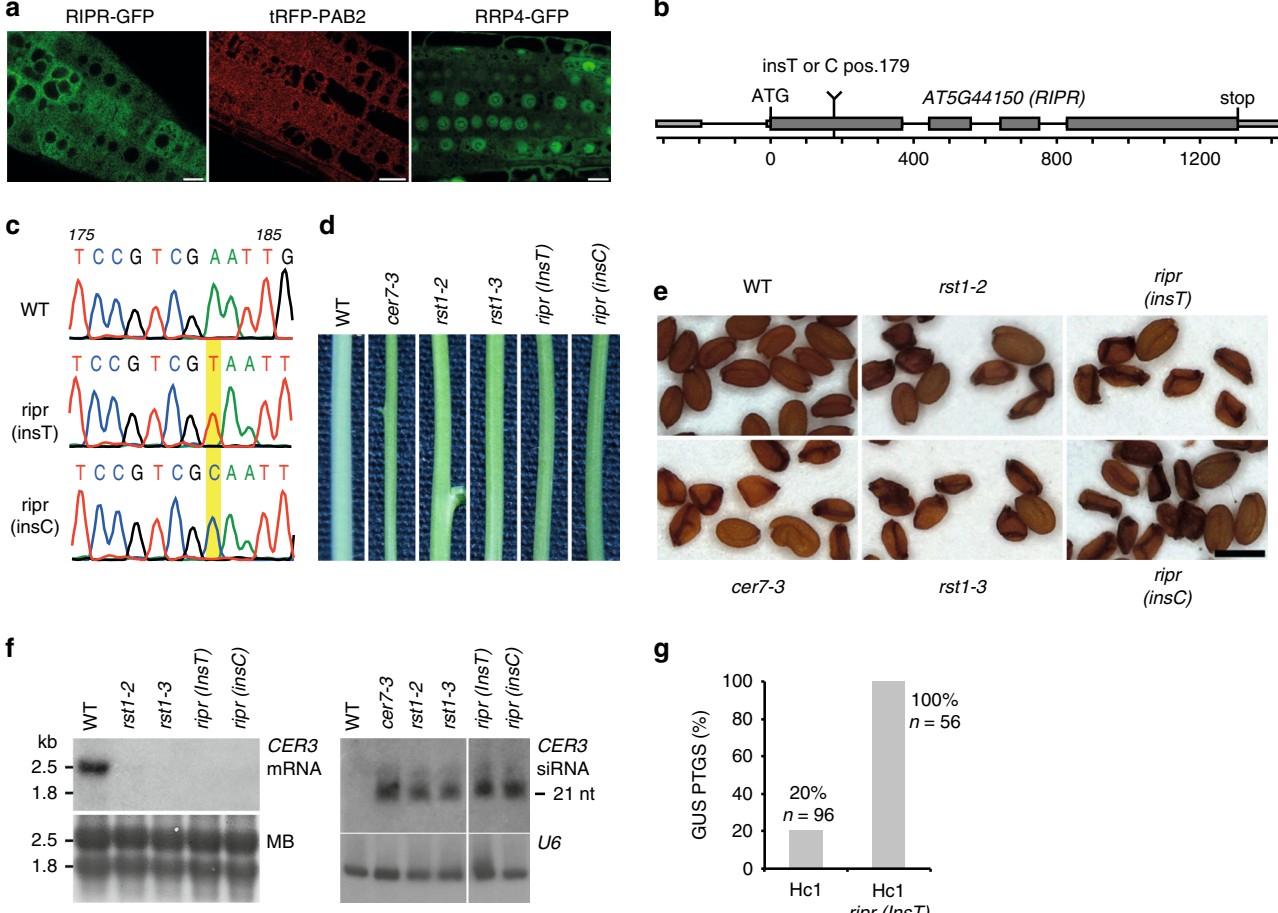

**Fig. 7** Loss of RIPR function phenocopies *rst1* mutants. **a** Confocal microscopy of plants expressing RIPR-GFP, the cytosolic marker tRFP-PAB2 and a GFP-tagged version of the exosome subunit RRP4. Scale bars are 20 μm. The source data are available at [https://doi.org/10.6084/m9.figshare.c.4483406]. **b** Scheme of the *AT5G44150* gene encoding the RIPR protein. Small and large boxes represent exons in the UTRs and CDS, respectively, lines represent introns. The single insertion of a T or C nucleotide at position 179 (from the ATG) results in a frameshift, which creates a premature stop codon. **c** Electropherograms of the genomic DNA sequence surrounding the relevant position of the *AT5G44150* locus in wild-type, *ripr(insT)* and *ripr(insC)* plants. **d** Stem sections from wild-type (WT), *cer7*, *rst1* and *ripr* plants. **e** *cer7-3*, *rst1* and *ripr* mutants produce similar proportions of non-viable seeds. Scale bar is 0.5 mm. Please also see the uncropped pictures provided in Supplementary Fig. 4. **f** Northern blots showing the downregulation of the *CER3* mRNA (left) and the upregulation of *CER3*-derived siRNAs (right) in *ripr* mutants. Loading controls show the methylene blue stained membrane (MB, left) and the hybridisation with a probe specific to *U6* snRNA (right). The uncropped blots can be seen in Supplementary Fig. 5, and are also available at [https://doi.org/10.6084/m9.figshare.c.4483406]. **g** RIPR is a silencing suppressor. The barplot shows the percentage of plants that spontaneously trigger silencing of the Hc1 35S prom:GUS S-PTGS reporter

exosome core complex and the Ski complex, respectively, both of which act together in the degradation of cytoplasmic RNAs. Secondly, both RST1 and RIPR suppress the silencing of transgenic reporters similar to almost all known main RNA degradation factors including proteins involved in decapping, the 5′−3′ exoribonucleases XRN3 and XRN4, and both subunits of Exo9 and Exo9 cofactors involved in 3′−5′ RNA degradation[25,55–57,59]. Thirdly, loss-of-function mutations of either RST1, RIPR or the exosome subunit CER7 lead to the accumulation of illegitimate siRNAs generated from endogenous protein-coding genes many of which have been previously found to produce siRNAs in *ski2*, *xrn4 ski2*, or in mutants of the decapping complex[57,58,60,66].

The current hypothesis for the production of rqc-siRNAs is that mRNA degradation intermediates, such as decapped, deadenylation or cleaved mRNAs, including the fragments produced by RISC, must be rapidly eliminated to avoid that they serve as substrates for the synthesis of double-stranded RNA by endogenous RNA-dependent RNA polymerases and SGS3[55–58,60,67]. Likely, the largely redundant functions of the

cytoplasmic 5′−3′ exoribonuclease XRN4, the 3′−5′ exoribonuclease DIS3L2/SOV and the cytoplasmic exosome together with the Ski complex ensure the rapid elimination of mRNAs after their degradation has been initiated by decapping, deadenylation or RISC-mediated cleavage, thus preventing the production of rqc-siRNAs. Vice versa, accumulation of rqc-siRNAs indicates impaired RNA degradation. The fact that similar rqc-siRNA profiles are observed in *cer7*, *rst1* and *ripr* mutants indicates that CER7, RST1 and RIPR contribute to the degradation of an overlapping set of mRNA targets.

Due to a natural variation, the Col-0 accession that is used as wild-type here and in most other studies investigating RNA degradation in plants lacks a fully functional DIS3L2/SOV 3′−5′ exoribonuclease and can therefore be regarded as a *sov* mutant[66,68]. Compared with plants expressing a functional SOV homologue, Col-0 does not accumulate rqc-siRNAs (except siR-NAs derived from the *AT2G01008* mRNA)[66], perhaps because most of its RNA substrates can also be degraded by the cyto-plasmic exosome. Therefore, we cannot exclude that the

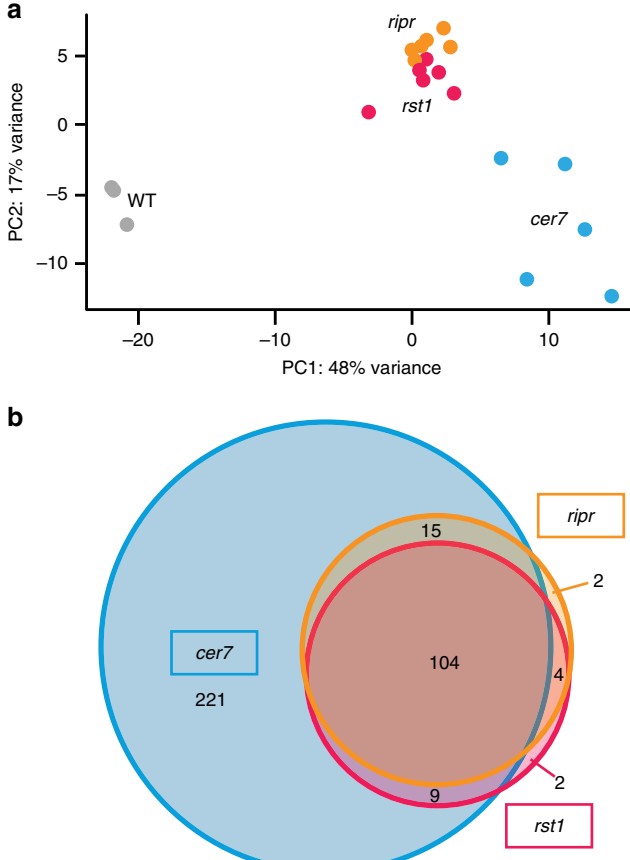

**Fig. 8** Loss of RST1 or RIPR results in the accumulation of small RNAs that are also produced in *cer7* mutants **a** Multidimensional scaling plot illustrating global variance and similarities between the 21/22 nt small RNA populations detected in the replicates of WT, *cer7*, *rst1* and *ripr*. **b** Venn diagram showing that *rst1* and *ripr* mutants accumulate quasi identical populations of small RNAs almost all of which are also detected in *cer7* mutants. The source data are available in Supplementary Data 5

accumulation of rqc-siRNAs in *cer7*, *rst1* and *ripr* is only observed because these mutants simultaneously lack SOV. However, the fact that the wax-deficient phenotype caused by the production of *CER3*-derived siRNAs is also observed in Landsberg and C24 accessions, both of which possess a functional SOV protein, implies that SOV and the cytoplasmic RNA exosome are not fully redundant.

It is important to note that both the loci concerned and the levels of siRNAs from a given loci vary among plants of the same genotype and grown under identical conditions. Not each of the loci that give rise to rqc-siRNA undergoes silencing, i.e., full suppression of its expression[66]. Yet, PTGS is obviously consistently triggered for certain *Arabidopsis* mRNAs. For instance, compromising 3′−5′ degradation by the exosome (and in the absence of SOV in the Col-0 accession) leads to silencing of the *CER3* and few other mRNAs, while distinct loci appear to be more sensitive to impaired decapping or 5′−3′ decay[57,58,60]. Interestingly, many of the mRNAs that reproducibly generate small RNAs in RNA degradation mutants, including the *CER3* mRNA, have actually no sequence homology to miRNAs (Supplementary Data 5). It is therefore tempting to speculate that mRNAs prone to PTGS possess common intrinsic features that trigger the recruitment of SGS3 and RDR6. One of the currently discussed propositions is that highly expressed mRNAs are more likely to generate aberrant RNAs than moderately or low

expressed ones[60]. In addition, certain mRNAs prone to RNA silencing may be cleaved by off-targeted RISC, or are perhaps substrates of other endonucleases. Alternatively, secondary structures or strong association to proteins may impede degradation by at least one of the otherwise largely redundant 5′−3′ and 3′−5′ degradation pathways and could explain why some mRNAs are more likely to become substrate for RNA-dependent RNA polymerases than mRNAs which are efficiently degraded from both directions. Yet, about 30% of the rqc-siRNAs generating loci in *rst1* and *ripr* are known or predicted miRNA targets (Supplementary Data 5). Hence, at least for those, the initial substrate for RDR6-dependent siRNA production could be a RISC-cleaved mRNA fragment. Since miRNAs and AGO1, the main effectors of RISC, are associated with polysomes[69,70], 5′ cleavage products that could be generated by RISC on polysomes resemble truncated mRNAs without a stop codon and without a polyA tail. Therefore, we can presume that RST1 and RIPR, together with the Ski complex and the RNA exosome, participate in the elimination of no-stop RNA. The notion that 5′ RISC-cleaved fragments that fail to be degraded by RST1–RIPR–SKI and the exosome become a substrate for the production of small RNAs fits well with the observation that the full-length cleavage fragments of only 10–20% of the *Arabidopsis* miRNA targets can be detected in the non-stop decay mutant *pelota*[67]. Of note, this study and ours used the Col-0 accession, which does not express a functional SOV/DIS3L2[68]. Therefore, the respective contribution of the exosome and the SOV/DIS3L2 pathways to prevent the production of siRNAs from 5′ fragments of RISC-cleaved mRNAs remains to be specifically addressed.

Our data also have important implications on the physical organisation of the cytoplasmic RNA exosome and the Ski complex. In *Arabidopsis* and closely related species, two genes encode the exosome core subunit RRP45. *Arabidopsis* RRP45A and RRP45B/CER7 share 88% identity over their first 300 amino acids. Both subunits are located in cytosolic and nuclear compartments, and are at least in stable transformants, similarly enriched in nucleoli. Interestingly, both RST1 and RIPR only associated with RRP45B/CER7-containing exosomes, while no peptide of RST1 was detected in any of the 14 experiments that we performed using RRP45A as bait. Instead, RRP45A-containing exosomes preferentially co-purified with the nuclear RNA helicase HEN2, in line with previous results obtained using HEN2 as bait[25]. Our results indicate that RST1 preferentially associates with the CER7-containing version of the *Arabidopsis* exosome. Compared with RRP45A, CER7 possesses an extra C-terminal domain of 135 amino acids, which may be important for the recruitment of RST1. However, a previous study suggested that these extra amino acids are dispensable for the function of CER7 in the degradation of the *CER3* mRNA[49]. Moreover, the ectopic expression of RRP45 in *cer7* mutants rescued their wax-deficient phenotype[49]. A possible explanation is that the *cer7-3* allele might be a knockdown rather than a knockout mutant and still express residual amounts of CER7, which may be sufficient to promote the degradation of *CER3* mRNAs when elevated levels of RRP45-containing exosomes take over other functions such as nuclear RNA surveillance. We can also speculate that overexpression of RRP45A allows a weak interaction with RST1 that is below the detection level in our IPs with RRP45A as bait. An alternative scenario could be that the physical interaction between RST1 and Exo9 is not essentially required for the function of both proteins in the turnover of the *CER3* mRNA.

The observation that RST1 is amongst the most enriched proteins captured with either RRP41 or CER7 as bait suggests that RST1 is associated with the exosome core complex. The strong enrichment of both RIPR and SKI7 with RST1 as bait and the

observation that the Ski complex purifies mainly with RIPR suggests that RST1 and RIPR link the exosome to the Ski complex in plants. Future experiments will address the possibility that RIPR may be required to link the Ski complex with the core exosome while RST1 could stabilise the binding of Exo9 and SKI7. Other interesting possibilities are that RST1 and/or RIPR affect the recognition of target RNAs or the recruitment of the exosome to ribosomes. Yet, we find only a few ribosomal proteins enriched in individual IPs (Supplementary Data 1–4, or explore the interactive volcano blots available on figshare [https://doi.org/10.6084/m9.figshare.c.4483406]. Hence, we do not detect the association of the exosome to the ribosome that was observed in yeast[34,35,45]. Whether this has a technical basis or truly reflects a poor association of Exo9–RST1–RIPR–Ski complex with ribosomes remains to be investigated.

Interestingly, a recent study in yeast identified Ska1 as a protein that impedes the association of the yeast Ski–Exo9 complex with the ribosome[45]. Similar to RIPR in *Arabidopsis*, Ska1 affinity captured the Ski complex. But unlike RST1 or RIPR, the Ska1–Ski complex is not required for the degradation of coding regions, and instead, has a specific function in the elimination of RNAs devoid of ribosomes such as 3′ UTRs or long non-coding RNAs. Apparently, overexpressing of Ska1 outcompetes the association of the Ski complex with ribosomes, suggesting that the association the Ski complex with either Ska1 or the ribosome is mutually exclusive[45]. Of note, sequence homologues of Ska1 seem to be restricted to *S. cerevisiae* and some closely related fungi, although proteins with similar functions may exist in other species.

RIPR seems to be conserved in flowering plants but is absent from the genomes of mosses and green algae, suggesting a relative recent evolutionary origin. By contrast, RST1 is deeply conserved in the green lineage. Moreover, a single ARM-repeat protein comprising the same domain of unknown function DUF3037 (IPR022542) as RST1 is conserved in humans, across all metazoa and in ancient amoebozoa such as *Dictyostelium*, but is apparently absent from modern fungi (PTHR16212 protein family). The human DUF3037 protein KIAA1797 was named Focadhesin, because its GFP fusion protein has been detected in focal adhesion points of astrocytoma cells[71]. Interestingly, a recently generated high-throughput data set monitoring the migration of proteins in sucrose gradients with or without RNase treatment detected Focadhesin as a putative component of an RNA-dependent complex[72] [http://r-deep.dkfz.de/]. More work is needed to fully understand the molecular function of human Focadhesin. It will be interesting to investigate whether Focadhesin is also associated with the function of the RNA exosome in animals.

## Methods

**Plant material**. Plants were grown on soil or in vitro on the Murashige and Skoog medium supplemented with 0.5% sucrose at 20 °C in 16 h light and 8 h darkness. All plants were of the Col-0 accession, which served as wild-type in all experiments. The T-DNA insertion lines *cer7-2* (Salk_003100), *cer7-3* (GK_089C02), *rrp45a* (GK_665D02), *sgs3-13* (Salk_039005) and *rst1-2* (Salk_070359), *rst1-3* (Salk_129280) have been described in refs. [49,50] and [47], respectively. The *rst1-5* and *rst1-4* alleles are EMS alleles identified during this study. Starting point for the identification of *rst-5* was the EMS mutagenesis of the line *L1 jmj14-4* line, in which PTGS of the *35Sprom:GUS* transgene inserted at the *L1* locus is partially impaired by the *jmj14-4* mutation[62]. The *rst1-4* mutant was identified following EMS treatment of *MIM156*[61]. EMS mutagenesis of seeds was performed as described in ref. [73]. Mutations were identified by mapping-by-sequencing using pooled F2 plants exhibiting the phenotype of interest. Sequencing libraries prepared with the Illumina TruSeq DNA Sample Preparation Kit were 10-plexed (Illumina adapters Set A) per flow-cell lane and sequenced on an Illumina HiSEquation 2000 instrument to obtain at least tenfold genome coverage. The SHOREmap technique was used to identify SNPs and mapping intervals. The EMS mutants were back-crossed to Col-0 to remove the *MIM156* transgene (rst1-4) or the *jmj14-4* mutation

and the *L1 35Sprom:GUS* reporter (rst1-5). Presence or absence of the transgenes and mutations were confirmed by PCR genotyping.

**PTGS analysis**. *Hc1 rst1-5*, *Hc1 ripr(insT)* and *6b4 rst1-5* plants were obtained by crosses. S-PTGS frequencies were assessed by GUS activity assays. Briefly, 0.5-1 µg of soluble proteins extracted from inflorescence leaves were incubated with 150 µl of 2 mM 4-methyl-umbelliferym-β-glucuronide, 29 mM $Na_2HPO_4$, 21 mM $NaH_2PO_4$, 10 mM EDTA. Fluorescence was determined at 15 s intervals for 30 min at 37 °C with a Fluoroscan Ascent 2.6. The GUS activity corresponds to the slope of the curve. Typically, GUS activity in non-silenced 6b4 and Hc1is >500 FLUO min$^{-1}$ µg$^{-1}$. Plants are considered silenced if GUS activity is <50 FLUO min$^{-1}$ µg$^{-1}$.

**CRISPR-Cas9 editing of *AT5G44150***. The target site at position + 179 from the ATG of the *AT5G44150* gene was selected using the CRISPR plant webtool [http://www.genome.arizona.edu/crispr/CRISPRsearch.html]. No off-targets were predicted for the guide RNA TCATACCGATCCCAATTCGA targeting the complementary strand at Chr5:17764907-17764927. Hundred picomoles of the oligonucleotides 5′-ATTGTCATACCGATCCCAATTCGA-3′ and 5′-AAACTCG AATTGGGATCGGTATGAC-3′ were phosphorylated for 30 min at 37 °C using 1 mM ATP and 1 unit polynucleotide kinase (NEB) in the buffer supplied by the manufacturer and then hybridised in a thermocycler (5 min 95 °C, cooling rate 5 °C/min, 5 min 25 °C). Hundred femtomoles of hybridised oligonucleotides were ligated overnight at 18 °C to 10 ng of Aar1-digested and dephosphorylated vector pKI1.1 R[74]. An aliquot of the ligation mixture was transformed in TOP10 *E. coli* cells (Invitrogen). The correct insertion of the guide RNA in the vector was confirmed by Sanger sequencing before plasmids were introduced in *Agrobacterium tumefaciens* strain GV3101 for the transformation of Col-0 plants by floral dip. pKI1.1R's T-DNA confers a red fluorescence protein expressed under the seed-specific *OLEO1* promoter. Fluorescent T1 seeds were selected using an epifluorescence-equipped binocular. Plants were genotyped by high-resolution melting using the precision melt supermix (Biorad) in a Roche Lightcyler 480 and further confirmed by Sanger sequencing. Plants carrying an insertion at the *AT5G44150* target site were selfed, and RFP-negative T2 seeds devoid of the Cas9-containing T-DNA were selected for outgrowth. Two independent T2 plants homozygous for the insertion of a single T or C at position + 179 were selected for further characterisation.

**Expression of GFP-tagged fusion proteins**. RRP41-GFP lines were made with constructs comprising the genomic sequence of *RRP41* including 1000 bp upstream of the translation start site and were previously described in ref. [10]. All other GFP fusion proteins were expressed from the *UBIQUITIN 10* promoter. C-terminal fusion constructs contained the genomic sequence of the respective genes, including the 5′ UTR, but lacking the Stop codon. For N-terminal fusions, the genomic sequences without the 5′ UTR, but including the 3′ UTR were used. All sequences were amplified from genomic DNA, cloned into pENTR1a (Invitrogen) and transferred to pUBC-GFP and pUBN-GFP destination vectors[75], respectively, using Gateway[R] recombinases. Expression vectors were transferred to *Agrobacterium* and used to transform rst1-3 (for RST1-GFP and GFP-RST1), cer7-2 for CER7-GFP, rrp45a for RRP45a-GFP and Col-0 plants for both GFP-RIPR and RIPR-GFP.

**Co-immunopurification experiments**. Plants were selected by testing crude flower extracts by western blots using homemade antibodies specific to GFP. For each IP, 200–500 mg of flower buds pooled from at least four individual plants were ground in liquid nitrogen or directly in 2 ml of ice-cold lysis buffer (50 mM Tris HCL pH 7.5, 25 or 50 mM NaCl, 1% Triton, protease inhibitors (Complete–EDTA, Roche). After removal of cell debris by centrifugation (two times 5 min, 16000×*g*, 4 °C) the cleared supernatants were incubated for 30 min with 50 µl of magnetic microbeads coupled to GFP antibodies (Miltenyi, catalogue number 130-091-125). Beads were loaded on magnetised MACS separation columns equilibrated with lysis buffer, and washed five times with 300 µl of washing buffer (50 mM Tris HCl pH 7.5, 25 or 50 mM NaCl, 0.1% Triton). Samples were eluted in 50 µl of pre-warmed elution buffer (Milteny). Control IPs were carried out with GFP antibodies in Col-0 or in plants expressing RFP or GFP alone. Additional control IPs were performed with antibodies directed against myc or HA epitopes (Miltenyi, catalogue numbers 130-091-123 and 130-091-122).

Eluted proteins were digested with sequencing-grade trypsin (Promega) and analysed by nanoLC-MS/MS on a QExactive + mass spectrometer coupled to an EASY-nanoLC-1000 (Thermo-Fisher Scientific, USA). The data were searched against the TAIR10 database with a decoy strategy. Peptides were identified with Mascot algorithm (version 2.5, Matrix Science, London, UK), and the data were imported into Proline 1.4 software [http://proline.profiproteomics.fr/]. The protein identification was validated using the following settings: Mascot pretty rank <= 1, FDR <= 1% for PSM scores, FDR <= 1% for protein set scores. The total number of MS/MS fragmentation spectra was used to quantify each protein from at least two independent biological replicates. If not specified otherwise, biological replicates consisted of plants of the same genotype grown at different dates and in different growth chambers.

For the statistical analysis of the co-immunoprecipitation data, we compared the data collected from multiple experiments for each bait against a set of 20 control IPs using R v3.5.1, R-studio v1.1.453. The size factor used to scale samples were calculated according to the DESeq2 normalisation method (i.e., median of ratios method)[76]. edgeR v3.14.0 and Stats v3.3.1 were used to perform a negative-binomial test and calculate the fold change and an adjusted $p$-value corrected by Benjamini–Hochberg for each identified protein. MDS plots were calculated with Stats v3.3.1. Annotation of proteins was retrieved using BiomartR v2.28.0, and volcano plots were drawn with ggplot2 v3.1.0. The RST1 data set comprised the data collected from 15 immunoprecipitation experiments performed in 6 biological replicates of each GFP-RST1 and RST1-GFP. Fourteen IPs from four biological replicates were performed with RRP45A. Six experiments from two biological replicates were used for each of RRP41-GFP and CER7-GFP, and the RIPR data set contained four IPs from two biological replicates of each RIPR-GFP and GFP-RIPR. Control IPs included four biological replicates of Col-0 incubated with GFP antibodies, six IPs from four biological replicates of GFP-expressing plants treated with GFP antibodies and ten IPs performed with anti-HA antibodies in three biological replicates RST1-GFP, three replicates GFP-RST1 and in 1 RFP sample. The mass spectrometry proteomics data have been deposited to the ProteomeXchange Consortium [http://proteomecentral.proteomexchange.org] via the PRIDE partner repository[77] with the data set identifier PXD013435.

**Epicuticular wax analysis**. For each sample, three stem sections of 6 cm were immersed for 30 s in 10 ml of chloroform. Extracts were dried under $N_2$ gas, dissolved in 150 μl of chloroform, transferred in an insert and again dried under $N_2$ gas. Extracts were derivatized in a mix of BSTFA [$N,O$ bis(trimethylsilyl) tri-fluoroacetamide] (> 99%, Sigma)]/pyridine (> 99.5%, Sigma) (50/50, V/V) (1 h at 80 °C with shaking at 300 rpm) before BSTFA-pyridine extracts were evaporated under $N_2$ gas. The samples were dissolved in chloroform containing a mix of nine alkanes (C10–C12–C15–C18–C19–C22–C28–C32–C36) as internal standards. Derivatized silylated samples were analysed by GC-MS (436-GC, Bruker; column 30-m, 0.25-mm, 0.25 μm; HP-5-MS) with He carrier gas inlet pressure programmed for constant flow of 1 ml/min and mass spectrometric detector (SCION TQ, Bruker; 70 eV, mass to charge ratio 50–800). GC was carried out with temperature-programmed injection at 50 °C over 2 min. The temperature was increased by 40 °C/min to 200 °C, held for 1 min at 200 °C, further increased by 3 °C/min to 320 °C and held for 15 min at 320 °C. Injector temperature was set to 230 °C with a split ratio of 3:1. Peaks in the chromatogram were identified based on their mass spectra and retention indices. Mass spectra detected by GC-MS were compared with the spectra of known compounds stored in the National Institute Standard and Technology (NIST) and in the Golm Metabolome databases. Nonacosane, 15-nonacosanone and 1-octacosanol were identified with match values of 933, 865 and 952, respectively. Mass spectrometric detector peak areas were used for relative quantification with octacosane as internal standard.

**RNA extraction and northern blots**. RNA was extracted from the top 3 cm of inflorescence stems or from flowers with TRI-reagent (MRC) following the manufacturers instructions. After precipitation with 0.8 vol isopropanol for 1–3 h at −80 °C, RNAs were collected by centrifugation (30 min 16,000 × $g$, 4 °C), washed twice with 70% EtOH, dissolved in water and further purified by adding 1 vol of phenol:chloroform:isoamylalcohol (25:24:1). The aqueous phase was transferred in a fresh tube, RNAs were precipitated overnight with 2.5 vol of EtOH at −80 °C, collected by centrifugation, washed twice with 70% EtOH and dissolved in water to ~2.5 μg/μl. For high-molecular-weight northern blots, 20 μg of the total RNA were separated in a 2% denaturing agarose gel containing 30 mM Tricine, 30 mM trie-thanolamine and 40 mM formaldehyde (4-6 h at 50 V). The RNA was blotted to Amersham Hybond-N + membranes (GE Healthcare Life Sciences) and UV-cross-linked (254 nm). Membranes were stained with methylene blue and hybridised to $^{32}$P radiolabelled DNA probes (DecaLabel, ThermoFischer) in PerfectHyb™ (Sigma) overnight at 65 °C. Sequences for primers used for amplification of probe templates are listed in Supplementary Table 1.

For low-molecular-weight and small RNA northern blots, 20 μg of the total RNA were separated in 5 and 17% polyacrylamide gels (19:1), respectively, containing 7 M urea in 100 mM Tris, 100 mM borate, 2 mM EDTA. RNA was transferred to Amersham Hybond-NX membranes (GE Healthcare Life Sciences) and either UV-cross-linked (LMW blots) or chemically cross-linked (small RNA blots) by incubation with 0.16 M l-ethyl-3-(3-dimethylaminopropyl) carbodiimide (EDC, Sigma) in 0.13 M 1-methylimidazole (Sigma), pH 8, for 1.5 h at 60 °C. Membranes were hybridised to radiolabelled DNA probes overnight at 45 °C. For loading controls, blots were stripped by boiling 0.1% SDS, and hybridised to a radiolabelled oligonucleotide specific to 7SL RNA, U6 snRNA or miR156. Oligonucleotide sequences are listed in Supplementary Table 1.

**Microscopy**. Plants were grown on MS agar plates supplemented with 0.5% sucrose. Roots from 10-day-old seedlings were excised, placed with water under a coverslip and examined with a ZEISS LSM 780 confocal microscope. The line expressing tRFP-PAB2 was a kind gift of C. Bousquet-Antonelli.

**small RNA libraries and analysis**. RNA was prepared as described above from flower buds of 6-week-old plants. For each genotype, three biological replicates of Col-0 (wild-type), *cer7-3*, *rst1-2*, *rst1-3*, *ripr(insT)*, *ripr(insC)* and two of *cer7-4* were grown at different dates in different growth chambers. Libraries were prepared from 1 μg of the total RNA using the NEB Next Multiplex Small RNA Library Prep Set for Illumina (NEB #E7300S and #E7580S) following the manufacturer's instructions. After ligation of primers, cDNA synthesis and PCR amplification, pooled libraries were loaded on a NOVEX 6% polyacrylamide gel with 100 mM Tris, 100 mM borate, 2 mM EDTA. Five fractions corresponding to 130–180 bp products were excised and eluted overnight in water. After EtOH precipitation, size and concentration of the fractions were checked with an Agilent 2100 Bioanalyzer (Agilent Technologies). The fraction of 140–150 bp containing the 21–22 nt small RNAs of interest was sequenced on a HiSeq 4000 sequencer (single-end mode 1 × 50 bp).

Sequence reads were trimmed from 3′-adapters and low-quality bases (q < 30) using cutadapt v1.18[78]. Reads were aligned without mismatches to the *Arabidopsis* TAIR10 genome using ShortStack v3.8.5[79] in unique mode (−u). Counts of 21 nt and 22 nt reads were extracted and annotated against TAIR10. Differential expression analysis was performed with DEseq2. The data obtained from the two alleles of *cer7*, *rst1* and *ripr* were analysed together. Downstream analysis and data visualisation were done with R. Only loci with log2FC > 1 and an adjusted $p$-value of < 0.01 were considered. PhasiRNA were identified with the help of ShortStack's phasing score (score >= 5). Potential miRNA target mRNAs were predicted using the psRNATarget webservice at [http://plantgrn.noble.org/psRNATarget][80].

**Statistical analysis**. For the statistical analysis of proteomic and sequencing data, we used negative-binomial models based on the edgeR and DEseq2 packages, respectively, which calculate the fold change and adjusted $p$-values with a two-sided Wald test.

**Gel and blot images**. Uncropped blots, gels and stem images are provided in Supplementary Fig. 5 and at [https://doi.org/10.6084/m9.figshare.c.4483406].

**Reporting summary**. Further information on research design is available in the Nature Research Reporting Summary linked to this article.

## Data availability

The small RNA-seq and mass spectrometry proteomics raw data that support the findings of this study have been deposited to the NCBI Gene Expression Omnibus (GEO) database, accession code GSE129736, and to the ProteomeXchange Consortium via the PRIDE[75] partner repository with the data set identifier PXD013435, respectively. Full resolution versions of all images, the wax analysis data, the processed small RNA-seq data and interactive volcano blots are available at figshare.

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

## Acknowledgements

We thank Nathalie Bouteiller for GUS assays, Ivan Le Masson for helping in the genetic screen that identified *rst1-5*, Sandrine Koechler for preparation of the small RNA library and Abdelmalek Alioua for high-resolution melting analysis. Special thanks to Hélène Zuber for the R-script that enabled the statistical analysis of co-IP experiments. This work has been published under the framework of the LABEX ANR-10-LABX-0036_NETRNA and ANR-17-EURE-0023 and benefits from a funding from the state managed by the French National Research Agency as part of the Investments for the future programme and the Centre National de la Recherche Scientifique. The funders had no role in study design, data collection and analysis, decision to publish or preparation of the paper.

## Author contributions

Study design: H.L. and D.G.; RNA analysis and co-purification experiments: H.L., S.Y.A. N., C.G.D., P.H. and C.P.; mass spectrometry: P.H., L.K. and J.C.; statistical analysis of co-IP data: H.L.; gas chromatography: J.Z.; analysis of small RNA libraries: D.P.; *L1 jmj14-4* suppressor screen, mapping of *rst1-5*, RNA analysis Fig. 2e, PTGS assays: N.B. and H.V.; MIM156 suppressor screen and mapping of *rst1-4*: M.C., E.S.K., P.L.M.L. and D.W.; writing: H.L. and D.G.; visualisation: H.L.; supervision: D.G. and H.L.; funding acquisition: D.G.
