## [Peer Review File · Nature Communications]

Reviewers' comments:

Reviewer #1 (Remarks to the Author):

Firstly, let me say that I cannot comment on the background plant cell and molecular biology - which I lack expertise and experience with. That said, I find this article to be both scientifically sound and interesting.

The field of RNA exosome research remains vigorous on account of the myriad RNA processing and turnover possibilities (many of the mechanisms of which are unaccounted for and/or undiscovered). Adapter proteins and complexes that regulate or modify these possibilities are the subject of a significant amount of research interest and effort - generally speaking, illuminating the complexities of post-transcriptional gene regulation and RNA metabolism. This manuscript fits squarely in the paradigm. Study of the alternative RRP45 (A/B) subunits, in addition to the RST1/RIPR adapters, is a mentionable strength of this manuscript.

I recommend publication with mostly minor revisions as follows:

PAGE 3, lines 31-33: "...RNAs either from 5' to 3' or from 3' to 5' also prevent that degradation intermediates such as uncapped or RISC-cleaved mRNAs are attacked by post-transcriptional gene silencing (PTGS)..."

 also prevent that... are attacked - reads unclear to me, can this be tidied up?

If I understood: the turnover of intermediates such as uncapped or RISC-cleaved mRNAs prevents them becoming substrates for, or even being encountered by, post-transcriptional gene silencing. Perhaps it can be reworded in this way?

PAGE 6, lines 20-22: "Therefore we conclude that RST1 is a cytoplasmic protein, which is in contrast to a previous study, which had proposed that RST1 is localized at the plasmamembrane59"

 seems to me that this discrepancy should be further discussed, and a possible rationalization for the distinction put forward (in discussion)? There are probably plenty of technical reasons for this difference - why is the current observation more believable - aside from the fact that it is what these authors observed on this occasion in their system.

PAGE 6, lines 29-30: "The lack of HEN2 and MTR4 detection is in agreement with the cytoplasmic localization of RST1."

 [MAJOR] failure to detect, especially in IP/MS, can arise from so many reasons. I caution against the use of this language in the results section - this is not a result. It is a supposition - perhaps a correct one, to be determined - and belongs in the discussion section. I have made a few other similar comments below.

PAGE 7, lines 3-5: "The three components of the cytoplasmic Ski complex were also detected in RST1 IPs, albeit their enrichment relative to control IPs did not pass the $p < 0.05$ threshold."

 [MAJOR] I cannot abide authors conveniently ignoring a significance threshold set by the authors themselves. Without writing a novel on the current schism in stats use/reporting in biology - it is not OK to ignore a threshold set by the authors when it is convenient to highlight that your favorite proteins "almost made it." The result indicates (or should, presuming the model used is good) that, at the threshold set, the proteins mentioned were not significantly enriched / could have been that enriched by chance at the frequency covered by the threshold. Hence, it is not a result that can be discussed affirmatively for their biology model in the results section. This should be removed. The author's are at liberty to arbitrarily set a less strict threshold, present that

transparently, and discuss accordingly. Or, to conduct an alternative, complementary or orthogonal experiment such as quantitative IP/western - the targeted nature of which, having only a single hypothesis, makes it much easier to reach standard significance thresholds.

PAGE 7, lines 22-23: "By contrast, RST1 and RIPR were not detected when RRP45A was used as bait, even though RRP45A and CER7 are both present in cytosolic and nuclear compartments..."

 again, there's many ways to fail to observe. Were the comparisons done in a quantitative way such that RRP45 and CER7 were at ~molar parity? Was accounting done on the IPs to ensure the fractions of RRP45 and CER7 extracted and bound to the resin were comparable and equivalent? Perhaps the fraction of RST1 and RIPR that bind RRP45A goes to the pellet under the same in vitro conditions where the CER7 bound fraction is soluble and co-IP-able? Etc etc. If these kinds of controls have been done - then they should be more explicitly presented - if not, this kind of statement/assertion/supposition should be moved to the discussion - it's not a result, per se, by my point of view.

PAGE 7, lines 25-26: "...was sufficient to detect the nuclear RNA helicase HEN2 and exoribonuclease RRP44, whose association to Exo9 is rather labile in Arabidopsis..."

 add reference

PAGE 7, lines 26-28: "These results indicate that RST1 and RIPR are bound to Exo9 with RRP45B/CER7 subunits, but not to Exo9 complexes that contain the alternative subunit RRP45A."

 please move to discussion.

PAGE 8, lines 3-4: "Furthermore, our data indicate that RIPR bridges the association of RST1-SKI7 with the Ski complex."

 please move to discussion.

PAGE 11, lines 20-22: "The observation that RST1 is among the most enriched proteins captured with either RRP41 or CER7 as bait suggests that RST1 is stably associated with the exosome core complex."

 [MAJOR] how do you rule out post-lysis binding of RST1 to the exosome? This interaction may be stable in vitro, may even be driven to the on state in vitro. I don't dispute that this interaction is valid - I dispute claims regarding its stability or lability based on IP/MS conducted in a single in vitro biochemical condition. Better support this with additional data / rationale or avoid making claims about stability if your intention is to imply in vivo stability as opposed to in vitro stability in your very particular handling conditions.

 indeed, you employ this nuanced perception when discussion ribosome associations on lines 30-31 of the same page - so, I suggest treating all such similar claims accordingly.

I look forward to seeing the revised manuscript in press.

Yours sincerely,
John LaCava

Reviewer #2 (Remarks to the Author):

The RNA exosome is an important protein complex; it is actually two complexes with largely overlapping subunit compositions that localize to the nucleus and cytoplasm, and where they carry out RNA trimming, cleavage, and decay. In contrast to its well known internal core complex (exo9), the SKI complex is known to function with exo9 in the cytoplasm, but full understanding of the components, and insight into mRNA substrates of the cytoplasmic exosome, and largely unknown. This manuscript advances the field by identifying RST1 and RIPR are components of the Arabidopsis cytoplasmic exosome, and demonstrating their probable positioning between the exo9 core and the ski complex. In addition, by characterizing the small RNAs that arose in cer7 (exo9 component), rst1, and ripr mutants, the authors infer that RIPR and RST1 are necessary for degradation of a subset of RNA exosome substrates. Moreover, because the RST1 and RIPR small RNAs includes a high proportion that correspond to the 5' fragment that is predicted to arise from miRNA cleavage, the authors infer a role for the exosome in non-stop decay.

This is a beautiful manuscript. The experiments are well presented, and even the coIP include an admirable number of replicates. It is well written, and the results represent an important advance in our understanding of the RNA exosome.

However, I have one general concern. The work carried out here uses an Arabidopsis accession that lacks a second cytoplasmic 3'-to-5' RNA decay pathway, called SOV or AtDIS3L2. This is not acknowledged anywhere in the manuscript, and yet the results really need to be considered in terms of their possible in the manuscript, and yet it affects several of the conclusions drawn by the authors. I do not think the authors need to carry out new work that includes this pathway, but the writing needs to account for the possibility that they are analyzing synthetic phenotypes, as each RNA exosome mutant described as a single mutant is actually a double mutant that also is missing this second RNA decay pathway.

An example of the problems that relate to this start with the first sentence of the introduction, which claims that the main 3'-5' exoribonucleolytic activity is provided by the RNA exosome. This statement cites no studies, and I know of none that directly compare contributions of exosome and DIS3L2, but the absence of knowledge of relative importance doesn't excuse this statement. The failure to acknowledge the Dsov/DIS3L2 pathway in Arabidopsis probably comes from the lack of an abnormal phenotype of the single mutant, and yet it was recently demonstrated that the wild type-like phenotype arises from an over-compensating feedback pathway, and inferred that SOV contributes to decay of a large fraction of seedling-expressed genes.

A second place where a potential role for SOV should be acknowledged is the identification of 5' cleavage fragments from miRNAs; the presented data demonstrates a synthetic defect, and cannot distinguish between the RNA exosome or SOV being the primary activity leading to the accumulation of the identified small RNAs.

Specific Comments:

1. Line 6 states that the major 3' to 5' RNA decay pathway is the RNA exosome, however plant genomes encode a functional DIS3L2 ribonuclease, and DIS3L2 is very important in other systems. I do not know of any study that has compared the relative importance of RNA exosome and DIS3L2, so this claim, which lacks a citation, is not substantiated.
2. In the second paragraph of the introduction, the authors obliquely refer to exosome targeting complexes. The authors should define this term more directly. (page 2, lines 17-22)
3. Page 3 refers to Dom 34 (PELOTA) and releasing of stalled ribosomes. This was difficult to follow – it is probably less important for the paper that it arose by a whole genome duplication, and instead (or in addition) the authors should explain that they are referring to no-go decay in a way that readers can understand. (page 3, lines lines 4-5)

4. The authors make a compelling case that siRNAs accumulate in RNA decay mutants. However, the study used an Arabidopsis accession that lacks a functional DIS3L2 pathway. So, the appearance of the siRNAs might be a synthetic phenotype. I don't think the authors need to repeat experiments using strains with SOV, this is a beautiful study and makes important contributions to the field. However, they should not ignore the other major 3' to 5' decay pathway, and acknowledge where their conclusions cannot distinguish between roles of these two cytoplasmic decay activities.

5. Figure 7E claims to show similar proportions of shriveled seeds in mutants, but there is no quantification, only photos of a selection of seeds. Some sort of count seems warranted.

Reviewer #3 (Remarks to the Author):

This manuscript by Lange et al. substantially advances our knowledge about the composition of plant exosomes. The authors use a combination of genetic, biochemical and transcriptomic techniques to demonstrate that in Arabidopsis, two previously only partially, or completely uncharacterized proteins mediate interaction of the Ski complex with the core exosome. This is an interesting variation on the theme, and the work also resolves somewhat confusing previous genetic observations. I am pleased to state that this work is of very high quality overall, with the caveat that I am not an expert in the "small RNA" field.

From my perspective, this paper needs very little extra work before publication. The data are sound, so are the conclusions, and it is well written. The manuscript starts with a competent and comprehensive description of prior knowledge, and then delves into a succession of experiments that link the previously identified RST1 gene and the newly identified RIPR gene to the exosome, both at the biochemical level and initially through its impact on small RNA processing. I have only two requests:

- regarding the *rst1-4* allele, the authors show in Figure 2C that CER3 siRNAs are less abundant in this allele than in other *rst1* alleles. Does this mean that unlike in those other alleles, full length CER3 mRNA can be detected in *rst1-4*? Could the authors please show CER3 full length mRNA Northern blots for the *rst1-4* allele in comparison to wild type and at least one other allele?

- Regarding RST1 localization, the authors mention that the cytoplasmic localization they observe contrasts with previously reported plasma membrane localization. Looking at Mang et al. 2009, Figure 10 B, I feel that the localization was misinterpreted in this publication, there is quite some intracellular signal and the signal in highly vacuolarized cells could have been misinterpreted. The authors should comment on this, and highlight that they used the same C-terminal fusion as in the previous paper (on top of the N-terminal fusion). However, Mang et al. show a restricted, specific (vascular and epidermal?) expression pattern for RST1. Here it looks ubiquitous. So did the authors use the native RST1 promoter here? This is not clear to me from the Methods and should also be clearly indicated in the Figure and legend. And then, are RST1/RIPR maybe only present in a subset of tissues? Please clarify, unless I overlooked.

page 11, line 12: "amino acids"

Answers to the reviewers

We thank all three reviewers for their positive and constructive comments. Following their recommendations, we have made several modifications in the Results and Discussion sections. Moreover, we included an additional RNA blot in Figure 2 to show the intermediate accumulation of the *CER3* mRNA in the *rst1-4* allele. We also provide a new supplementary figure to better document the presence of shriveled seeds in both *rip1* mutants. Finally, the presentation of relative amounts of major stem wax compounds in Figure 1B was modified to show individual data point for the three replicates. The reviewer's link to access the raw data at Figshare is <https://figshare.com/s/67a45b45e45ce72e8d2e>

Reviewer #1 (Remarks to the Author):

Firstly, let me say that I cannot comment on the background plant cell and molecular biology - which I lack expertise and experience with. That said, I find this article to be both scientifically sound and interesting.

The field of RNA exosome research remains vigorous on account of the myriad RNA processing and turnover possibilities (many of the mechanisms of which are unaccounted for and/or undiscovered). Adapter proteins and complexes that regulate or modify these possibilities are the subject of a significant amount of research interest and effort - generally speaking, illuminating the complexities of post-transcriptional gene regulation and RNA metabolism. This manuscript fits squarely in the paradigm. Study of the alternative RRP45 (A/B) subunits, in addition to the RST1/RIPR adapters, is a mentionable strength of this manuscript.

I recommend publication with mostly minor revisions as follows:

PAGE 3, lines 31-33: "...RNAs either from 5' to 3' or from 3' to 5' also prevent that degradation intermediates such as uncapped or RISC-cleaved mRNAs are attacked by post-transcriptional gene silencing (PTGS)..."

 also prevent that... are attacked - reads unclear to me, can this be tidied up?

If I understood: the turnover of intermediates such as uncapped or RISC-cleaved mRNAs prevents them becoming substrates for, or even being encountered by, post-transcriptional gene silencing. Perhaps it can be reworded in this way?

Answer. The sentence was changed to:

"... the elimination of degradation intermediates such as uncapped or RISC-cleaved mRNAs by 3'-5' and 5'-3' exonucleases prevents that they trigger post-transcriptional silencing (PTGS) a mechanism required for the destruction of non-self RNAs originating from viruses or transgenes"

PAGE 6, lines 20-22: "Therefore we conclude that RST1 is a cytoplasmic protein, which is in contrast to a previous study, which had proposed that RST1 is localized at the plasmamembrane59"

 seems to me that this discrepancy should be further discussed, and a possible rationalization for the distinction put forward (in discussion)? There are probably plenty of technical reasons for this difference - why is the current observation more believable - aside from the fact that it is what these authors observed on this occasion in their system.

Answer. Please note that this concern is also discussed in an answer to Reviewer 3.

We share the opinion of Reviewer 3 that the membrane localization proposed in Mang et al. 2009 is a misinterpretation. These authors show a section of the upper elongation zone of roots, which contains fully differentiated cells. In those cells, most of the internal volume is filled by the large vacuole while the cytoplasm is restricted to a thin layer close to the plasma membrane. Therefore it is not the best choice to reliably discriminate proteins located in the

cytosol or at the plasma membrane. In addition, the single picture shown is of poor quality, maybe due to technical limitations. The interpretation of the published picture is further hampered by the presence of large spots, maybe aggregates or technical artifacts, which outshine the rather weak signal of RST1-GFP1.

In our manuscript, we provide both phenotypic and molecular evidences that the GFP-RST1 and RST1-GFP complement the *rst1* mutations. We also show high quality microscopy pictures and a comparison with a known cytosolic marker to support our conclusion that RST1 is a cytosolic protein. Moreover, the cytosolic pattern reported in our study was observed in several independent transformants, and with both C- and N-terminal tags upon stable expression in *Arabidopsis*. We contacted the Jenks laboratory a couple of years ago but unfortunately, the biological material published in Mang et al. was discarded. Because we could not perform a side-by-side comparison of the respective transgenic lines, we chose not to discuss this issue further in the manuscript. However, we think that it is both necessary and fair to alert readers that our localization data are in contrast to a former claim. Therefore we prefer to keep the following statement: "we conclude that RST1 is a cytoplasmic protein. This result is in contrast to a previous study, which had proposed that RST1 is localized at the plasma membrane". The alternative option, which is not the one we favour, is to state simply "we conclude that RST1 is a cytoplasmic protein".

PAGE 6, lines 29-30: "The lack of HEN2 and MTR4 detection is in agreement with the cytoplasmic localization of RST1."

 [MAJOR] failure to detect, especially in IP/MS, can arise from so many reasons. I caution against the use of this language in the results section - this is not a result. It is a supposition - perhaps a correct one, to be determined - and belongs in the discussion section. I have made a few other similar comments below.

Answer. We agree. The sentence was removed. We also removed a similar note from the figure legend.

PAGE 7, lines 3-5: "The three components of the cytoplasmic Ski complex were also detected in RST1 IPs, albeit their enrichment relative to control IPs did not pass the $p < 0.05$ threshold.

 [MAJOR] I cannot abide authors conveniently ignoring a significance threshold set by the authors themselves. Without writing a novel on the current schism in stats use/reporting in biology - it is not OK to ignore a threshold set by the authors when it is convenient to highlight that your favorite proteins "almost made it." The result indicates (or should, presuming the model used is good) that, at the threshold set, the proteins mentioned were not significantly enriched / could have been that enriched by chance at the frequency covered by the threshold. Hence, it is not a result that can be discussed affirmatively for their biology model in the results section. This should be removed. The author's are at liberty to arbitrarily set a less strict threshold, present that transparently, and discuss accordingly. Or, to conduct an alternative, complementary or orthogonal experiment such as quantitative IP/western - the targeted nature of which, having only a single hypothesis, makes it much easier to reach standard significance thresholds.

Answer. Our intention was to alert the readers that these proteins that are of interest in the context of this study were not significantly enriched. We did not ignore the threshold set to $\text{adj}P = 0.05$ as we explicitly wrote that the adjusted P-values were above.

To exclude any misunderstanding, we changed the text to:

"The three proteins of the SKI complex were not enriched in RST1 IPs (Fig. 4)". Furthermore, the phrase "The dashed line indicates the threshold above which proteins are significantly enriched ($\text{adj}p < 0.05$)." was added to the figure legends.

Please note that the next three comments on page 7 are addressed in a common answer below.

PAGE 7, lines 22-23: "By contrast, RST1 and RIPR were not detected when RRP45A was used as bait, even though RRP45A and CER7 are both present in cytosolic and nuclear compartments..."

 again, there's many ways to fail to observe. Were the comparisons done in a quantitative way such that RRP45 and CER7 were at ~molar parity? Was accounting done on the IPs to ensure the fractions of RRP45 and CER7 extracted and bound to the resin were comparable and equivalent? Perhaps the fraction of RST1 and RIPR that bind RRP45A goes to the pellet under the same in vitro conditions where the CER7 bound fraction is soluble and co-IP-able? Etc etc. If these kinds of controls have been done - then they should be more explicitly presented - if not, this kind of statement/assertion/supposition should be moved to the discussion - it's not a result, per se, by my point of view.

PAGE 7, lines 25-26: "...was sufficient to detect the nuclear RNA helicase HEN2 and exoribonuclease RRP44, whose association to Exo9 is rather labile in *Arabidopsis*..."

 add reference

PAGE 7, lines 26-28: "These results indicate that RST1 and RIPR are bound to Exo9 with RRP45B/CER7 subunits, but not to Exo9 complexes that contain the alternative subunit RRP45A."

 please move to discussion.

Answer. We agree that we cannot formally exclude that the failure to detect RST1 with RRP45A could have a technical explanation. We modify this part to describe the RRP45A and CER7 IPs with a descriptive and careful wording as follows:

"...both RST1 and SKI7 were significantly enriched with CER7 as bait (Fig. 5, Supplementary Table 3). RRP45A co-purified with the nucleoplasmic RNA helicase HEN2, which confirmed the association of HEN2 with Exo9-RRP45A previously observed in the reciprocal IP (Lange et al 2014). The high number of experiments (15 IPs) was also sufficient to detect the exoribonuclease RRP44, which is consistently poorly enriched in IPs of the *Arabidopsis* Exo9 (Fig. 4, Lange et al 2014, Sikorska et al 2017). We cannot formally rule out that the failure to detect RST1 and SKI7 with RRP45A-GFP is due to technical reasons. However, our data suggest that RST1 and SKI7 are principally associated with RRP45B/CER7. The difference in the "interactome" of CER7 and RRP45A cannot be explained by different intracellular localization of the baits, because both CER7-GFP and RRP45A-GFP were present in nuclear and cytoplasmic compartments (Supplementary Fig. 2)."

PAGE 8, lines 3-4: "Furthermore, our data indicate that RIPR bridges the association of RST1-SKI7 with the Ski complex."

 please move to discussion.

Answer. The sentence was changed to:

"Furthermore, our data suggest that RST1 binds Exo9 and SKI7, while RIPR binds to RST1-SKI7 and the Ski complex".

PAGE 11, lines 20-22: "The observation that RST1 is among the most enriched proteins captured with either RRP41 or CER7 as bait suggests that RST1 is stably associated with the exosome core complex."

 [MAJOR] how do you rule out post-lysis binding of RST1 to the exosome? This interaction may be stable in vitro, may even be driven to the on state in vitro. I don't dispute that this interaction is valid - I dispute claims regarding its stability or lability based on IP/MS conducted in a single in vitro biochemical condition. Better support this with additional data / rationale or avoid making claims about stability if your intention is to imply in vivo stability as opposed to in vitro stability in your very particular handling conditions.

Answer. We see the point. We removed the adverb ‘stably”

"The observation that RST1 is among the most enriched proteins captured with either RRP41 or CER7 as bait suggests that RST1 is associated with the exosome core complex."

.  indeed, you employ this nuanced perception when discussion ribosome associations on lines 30-31 of the same page - so, I suggest treating all such similar claims accordingly.

Answer. We apologize for the misleading wording. Some ribosomal proteins are enriched in individual IP, but most ribosomal proteins are below the significance threshold. Moreover, no ribosomal proteins are consistently detected in different IPs, nor do we see a preferential enrichment of 40S or 60S proteins. Our conclusion was therefore that the binding of the exosome to the ribosome that is observed in other species cannot be inferred from our data. To make this idea clearer, the text was changed to:

"Other interesting possibilities are that RST1 and/or RIPR affect the recognition of target RNAs or the recruitment of the exosome to ribosomes. Yet, we find only a few ribosomal proteins enriched in individual IPs (Supplementary Tables 1-4, or explore the interactive volcano blots available on figshare (<https://doi.org/10.6084/m9.figshare.c.4483406>). Hence, we do not detect the association of the exosome to the ribosome that was observed in yeast (Halbach 2013, Schmidt 2016, Zhang 2019). Whether this has a technical basis or truly reflects a poor association of Exo9-RST1-RIPR-Ski complex with ribosomes remains to be investigated"

I look forward to seeing the revised manuscript in press.

Yours sincerely,

John LaCava

We thank you for signing your review.

Reviewer #2 (Remarks to the Author):

The RNA exosome is an important protein complex; it is actually two complexes with largely overlapping subunit compositions that localize to the nucleus and cytoplasm, and where they carry out RNA trimming, cleavage, and decay. In contrast to its well known internal core complex (exo9), the SKI complex is known to function with exo9 in the cytoplasm, but full understanding of the components, and insight into mRNA substrates of the cytoplasmic exosome, and largely unknown. This manuscript advances the field by identifying RST1 and RIPR are components of the Arabidopsis cytoplasmic exosome, and demonstrating their probable positioning between the exo9 core and the ski complex. In addition, by characterizing the small RNAs that arose in cer7 (exo9 component), rst1, and ripr mutants, the authors infer that RIPR and RST1 are necessary for degradation of a subset of RNA exosome substrates. Moreover, because the RST1 and RIPR small RNAs includes a high proportion that correspond to the 5' fragment that is predicted to arise from miRNA cleavage, the authors infer a role for the exosome in non-stop decay.

This is a beautiful manuscript. The experiments are well presented, and even the coIP include an admirable number of replicates. It is well written, and the results represent an important advance in our understanding of the RNA exosome.

However, I have one general concern. The work carried out here uses an Arabidopsis accession that lacks a second cytoplasmic 3'-to-5' RNA decay pathway, called SOV or AtDIS3L2. This is not acknowledged anywhere in the manuscript, and yet the results really need to be considered in terms of their possible in the manuscript, and yet it affects several of the conclusions drawn by the authors. I do not think the authors need to carry out new work that includes this pathway, but the writing needs to account for the possibility that they are analyzing synthetic phenotypes, as each RNA exosome mutant

described as a single mutant is actually a double mutant that also is missing this second RNA decay pathway.

An example of the problems that relate to this start with the first sentence of the introduction, which claims that the main 3'-5' exoribonucleolytic activity is provided by the RNA exosome. This statement cites no studies, and I know of none that directly compare contributions of exosome and DIS3L2, but the absence of knowledge of relative importance doesn't excuse this statement. The failure to acknowledge the Dsov/DIS3L2 pathway in Arabidopsis probably comes from the lack of an abnormal phenotype of the single mutant, and yet it was recently demonstrated that the wild type-like phenotype arises from an over-compensating feedback pathway, and inferred that SOV contributes to decay of a large fraction of seedling-expressed genes.

RNA exosome or SOV being the primary activity leading to the accumulation of the identified small RNAs.

A second place where a potential role for SOV should be acknowledged is the identification of 5' cleavage fragments from miRNAs; the presented data demonstrates a synthetic defect, and cannot distinguish between the RNA exosome or SOV being the primary activity leading to the accumulation of the identified small RNAs.

Answer. We agree with this general comment of Reviewer 2. We made several text changes to acknowledge the role of DIS3L2/SOV as detailed below

Specific Comments:

1. Line 6 states that the major 3' to 5' RNA decay pathway is the RNA exosome, however plant genomes encode a functional DIS3L2 ribonuclease, and DIS3L2 is very important in other systems. I do not know of any study that has compared the relative importance of RNA exosome and DIS3L2, so this claim, which lacks a citation, is not substantiated.

Answer. The importance of the RNA exosome is often mentioned because mutations disrupting the core of the RNA exosome are lethal in all eukaryotes investigated to date. However, we agree that the respective importance of 3'-5' exoribonucleolytic activities should not be ranked, and that the DIS3L2/SOV pathway must be discussed in the manuscript. Because our manuscript is focused on the RNA exosome, we prefer to start the introduction directly with the exosome, and we will refer to the Dis3L2/SOV pathway in the discussion. We have changed "The main 3'- 5' exoribonucleolytic activity in eukaryotic cells is provided by the RNA exosome." by "The RNA exosome provides all eukaryotic cells with a key 3'- 5' exoribonucleolytic activity that participates in the maturation of various non-coding RNAs and in the degradation of both non-coding and coding RNAs (reviewed in e.g. Schaeffer et al. 2011; Lange and Gagliardi 2011; Tollervey 2015; Łabno, Tomecki, and Dziembowski 2016; Zinder and Lima 2017)".

2. In the second paragraph of the introduction, the authors obliquely refer to exosome targeting complexes. The authors should define this term more directly. (page 2, lines 17-22)

Answer. The sentence was changed to:

"In all eukaryotes investigated, the catalytic activities of the RNA exosome are modulated by cofactors termed activator-adaptor or exosome targeting complexes. These complexes aid in the recognition of specific types of RNA substrates and couple exosome-mediated degradation to cellular processes such as ribosome biogenesis or mitosis. All exosome targeting complexes ..."

3. Page 3 refers to Dom 34 (PELOTA) and releasing of stalled ribosomes. This was difficult to follow – it is probably less important for the paper that it arose by a whole genome duplication, and instead (or in addition) the authors should explain that they are referring to no-go decay in a way that readers can understand. (page 3, lines lines 4-5)

Answer. The first important point here is that the closely related proteins SKI7 and HBS1 are produced from duplicated genes in yeast but spliced from a single gene in mammals and plants. The second important point is that HBS1 can associate with the ribosome, while such a function was not demonstrated for SKI7 *in vivo*.

We changed the text to:

" Recent data revealed the functional conservation of SKI7 across eukaryotes (Marshall et al. 2018). In mammals and plants, SKI7 is produced by alternative splicing from a single locus that encodes also the HBS1 protein (Kalisiak et al. 2017; Marshall et al. 2018; Brunkard and Baker 2018). HBS1 functions together with the G-protein Dom34/PELOTA in No-Stop decay by releasing ribosomes stalled on RNAs lacking a stop codon (Doma and Parker 2006; Atkinson, Baldauf, and Haurlyliuk 2008; Tsuboi et al. 2012; Saito, Hosoda, and Hoshino 2013). In the yeast *Saccharomyces cerevisiae*, Ski7 and Hbs1 are closely related paralogs. Therefore, it was for long inferred that yeast Ski7 mediates the association of the exosome with the ribosome. Recent data now challenge this view by showing that the Ski2-Ski3-Ski8 complex can directly bind to ribosomes while Ski7 is associated with Exo9 (Schmid et al 2016, Zhang et al 2019).

4. The authors make a compelling case that siRNAs accumulate in RNA decay mutants. However, the study used an Arabidopsis accession that lacks a functional DIS3L2 pathway. So, the appearance of the siRNAs might be a synthetic phenotype. I don't think the authors need to repeat experiments using strains with SOV, this is a beautiful study and makes important contributions to the field. However, they should not ignore the other major 3' to 5' decay pathway, and acknowledge where their conclusions cannot distinguish between roles of these two cytoplasmic decay activities.

Answer. The wax-deficient phenotypes caused by point mutations in CER7 and RST1 were originally discovered in Landsberg and C24, both of which possess a functional SOV (Zhang 2010). Hence, the absence of functional SOV in the Col-0 accession is unlikely the reason for the accumulation of CER3-derived siRNAs in *rst1* or *cer7* (although we cannot formally exclude that the two original EMS mutants *cer7-1* and *rst1-1* are also simultaneously mutated in SOV). In any case, we do not rule out that some of the mRNAs that generate siRNAs in *cer7*, *rst1* or *ripr* are also targets of SOV in Landsberg and other accessions. We added a new paragraph addressing this possibility to the discussion:

"Due to a natural variation, the Col-0 accession that is used as wild type here and in most other studies investigating RNA degradation in plants lacks a fully functional DIS3L2/SOV 3'-5' exoribonuclease and can therefore be regarded as a *sov* mutant (Zhang 2010). Compared to plants expressing a functional SOV homologue, Col-0 does not accumulate *rqc*-siRNAs (except siRNAs derived from the *AT2G01008* mRNA (Sorenson et al. 2018)), perhaps because most of its RNA substrates can also be degraded by the cytoplasmic exosome. Therefore, we cannot exclude that the accumulation of *rqc*-siRNAs in *cer7*, *rst1* and *ripr* is only observed because these mutants simultaneously lack SOV. However, the fact that the wax-deficient phenotype caused by the production of CER3-derived siRNAs is also observed in Landsberg and C24 accessions, both of which possess a functional SOV protein, implies that SOV and the cytoplasmic RNA exosome are not fully redundant".

Furthermore, we added: "Of note, this study and ours used the Col-0 accession, which does not express a functional SOV/DIS3L2. Therefore, the respective contribution of the exosome and the SOV/DIS3L2 pathways to prevent the production of siRNAs from 5' fragments of RISC-cleaved mRNAs remains to be specifically addressed."

5. Figure 7E claims to show similar proportions of shriveled seeds in mutants, but there is no quantification, only photos of a selection of seeds. Some sort of count seems warranted.

Answer. The uncropped pictures for the seeds shown in Fig 7E were deposited at Figshare (<https://doi.org/10.6084/m9.figshare.c.4483406>), but we forgot to include them in the Supplementary Figure containing the raw pictures. We should have explicitly referred to these uncropped pictures in the legend of Fig 7, because they are indeed necessary to appreciate the presence of shriveled seeds in the different genotypes.

The uncropped pictures of seeds are now shown in a new dedicated Supplementary Figure S4.

Our point is that both independent *rip*r mutants produce shriveled seeds, as previously reported for *rst1*. We would like to communicate this observation without quantifying the respective amounts of shriveled seeds because we do not yet have the necessary number of independent seed stocks (harvested at different times) that could serve as biological replicates for a sound quantification. We also do not think that quantification is a key point here. The observation of shrunken seeds is rather descriptive, is not further experimentally addressed and is not important for the conclusions. Hence, an alternative solution to showing the uncropped pictures as Supplementary Figure S4 is to remove this information from the manuscript.

Reviewer #3 (Remarks to the Author):

This manuscript by Lange et al. substantially advances our knowledge about the composition of plant exosomes. The authors use a combination of genetic, biochemical and transcriptomic techniques to demonstrate that in Arabidopsis, two previously only partially, or completely uncharacterized proteins mediate interaction of the Ski complex with the core exosome. This is an interesting variation on the theme, and the work also resolves somewhat confusing previous genetic observations. I am pleased to state that this work is of very high quality overall, with the caveat that I am not an expert in the "small RNA" field.

From my perspective, this paper needs very little extra work before publication. The data are sound, so are the conclusions, and it is well written. The manuscript starts with a competent and comprehensive description of prior knowledge, and then delves into a succession of experiments that link the previously identified RST1 gene and the newly identified RIPR gene to the exosome, both at the biochemical level and initially through its impact on small RNA processing. I have only two requests:

- regarding the *rst1-4* allele, the authors show in Figure 2C that CER3 siRNAs are less abundant in this allele than in other *rst1* alleles. Does this mean that unlike in those other alleles, full length CER3 mRNA can be detected in *rst1-4*? Could the authors please show CER3 full length mRNA Northern for the *rst1-4* allele in comparison to wild type and at least one other allele?

Answer. An RNA blot showing the steady-state levels of *CER3* mRNAs in wild type, *cer7-3*, and all four *rst1* alleles used in this study is now shown in Fig. 2C. This blot shows that residual levels of *CER3* mRNAs are present in *rst1-4*. This result indicates that the *CER3* mRNA is incompletely silenced in this mutant, which is consistent with the low but detectable levels of *CER3*-derived siRNA. The new data strengthen our conclusion that *rst1-4* is a weaker allele than the alleles *rst1-2*, *rst1-3* and *rst1-5*.

- Regarding RST1 localization, the authors mention that the cytoplasmic localization they observe contrasts with previously reported plasma membrane localization. Looking at Mang et al. 2009, Figure 10 B, I feel that the localization was misinterpreted in this publication, there is quite some intracellular signal and the signal in highly vacuolarized cells could have been misinterpreted. The authors should comment on this, and highlight that they used the same C-terminal fusion as in the previous paper (on top of the N-terminal fusion). However, Mang et al. show a restricted, specific (vascular and epidermal?) expression pattern for RST1. Here it looks ubiquitous. So did the authors use the native RST1 promoter here? This is not clear to me from the Methods and should also be clearly indicated in

the Figure and legend. And then, are RST1/RIPR maybe only present in a subset of tissues? Please clarify, unless I overlooked.

Answer. Please note that the concern on the discrepancy on RST1 localization between our study and that of Mang et al. 2009 is also discussed in an answer to Reviewer 1.

We agree with Reviewer 3 that the localization of RST1 published in Mang et al. is likely a misinterpretation. We think that the cytosolic localization of RST1 as reported here is more reliable for the following reasons: the technical quality of the images are better; the cytosolic localization of RST1 fusion proteins is observed for both N- and C-terminal fusion proteins; the localization is consistent between independent transgenic lines and in all cells investigated; identical patterns are observed for RST1 fusion proteins and TagRFP-PAB2, a known cytosolic protein. Mang et al. used the 35SCaMV promoter to drive the expression of their transgene. Because 35SCaMV DNA sequences are present in many T-DNA insertion lines such as *rst1-2* and *rst1-3*, this promoter is often transcriptionally silenced (e.g. Mlotshwa et al 2010). Transcriptional silencing of the transgene may also explain the restriction of their signal to a few cells.

We used the *UBIQUITIN 10* promoter to drive the expression of RST1, CER7, RRP45 and RIPR fusion proteins. This information was indicated in the Methods under the header Expression of GFP-tagged fusion proteins: "All other GFP-fusion proteins were expressed from the *UBIQUITIN 10* promoter". To improve the visibility of this information, we have modified the legends of Figures 3, 7 and S2.

We would prefer not to modify the main text on the discrepancy on RST1 localization for the following reasons: We contacted the Jenks laboratory a couple of years ago to obtain the constructs and the lines published in Mang et al., but unfortunately all this material was discarded. The lack of this material prevents a side-by-side comparison with our transgenic lines. In the absence of such direct comparison, we prefer not to criticize the quality of the localization experiments published in Mang et al. We still need to inform readers by mentioning that our respective results are not in agreement. Hence, we kept the information: "we conclude that RST1 is a cytoplasmic protein. This is in contrast to a previous study, which had proposed that RST1 is localized at the plasma membrane". The alternative option, which is not the one we favour, is to state simply "we conclude that RST1 is a cytoplasmic protein".

page 11, line 12: "amino acids"

Corrected

REVIEWERS' COMMENTS:

Reviewer #1 (Remarks to the Author):

The authors competently and satisfactorily addressed my concerns by changes to the text. They appear to have similarly handled comments from the other reviewers, who were overall enthusiastic about the scientific quality of the original submission of this work.

I therefore now recommend publication of the revised manuscript.

Sincerely,
John LaCava

Reviewer #2 (Remarks to the Author):

The initially submitted version of this manuscript was already of very high quality. In the revision, the authors addressed all the issues I raised. I have no concerns with this version.

Reviewer #3 (Remarks to the Author):

The authors have comprehensively addressed my comments. Ready to publish from my side.